# Sfrp3 modulates stromal–epithelial crosstalk during mammary gland development by regulating Wnt levels

Ilenia Bernascone[1], Tamara González[1], Maria D. Barea[1], Claudia Carabaña[1], Mariam Hachimi[1], Minerva Bosch-Fortea[1], Silvia Santamaria[2], Raquel Martin[1], Julia Tarnick[3], Jose A. Garcia-Sanz [2] & Fernando Martín-Belmonte[1]

Mammary stroma is essential for epithelial morphogenesis and development. Indeed, postnatal mammary gland (MG) development is controlled locally by the repetitive and bidirectional cross-talk between the epithelial and the stromal compartment. However, the signalling pathways involved in stromal–epithelial communication are not entirely understood. Here, we identify Sfrp3 as a mediator of the stromal–epithelial communication that is required for normal mouse MG development. Using *Drosophila* wing imaginal disc, we demonstrate that Sfrp3 functions as an extracellular transporter of Wnts that facilitates their diffusion, and thus, their levels in the boundaries of different compartments. Indeed, loss of Sfrp3 in mice leads to an increase of ductal invasion and branching mirroring an early pregnancy state. Finally, we observe that loss of Sfrp3 predisposes for invasive breast cancer. Altogether, our study shows that Sfrp3 controls MG morphogenesis by modulating the stromal-epithelial cross-talk during pubertal development.

[1] Department of Development and Regeneration, Centro de Biologia Molecular Severo Ochoa (CSIC-UAM), 28049 Madrid, Spain. [2] Department of Molecular Biomedicine, Centro de Investigaciones Biologicas (CSIC), 28040 Madrid, Spain. [3] Centre for Integrative Physiology, University of Edinburgh, Edinburgh EH8 9XB, UK. Correspondence and requests for materials should be addressed to F.M.-B. (email: fmartin@cbm.csic.es)

The mammary gland (MG) is one of the few organs that undergoes most of its development postnatally. At puberty, the ducts of the rudimentary embryonic tree start to invade the mammary fat pad through a process of ductal elongation and bifurcation, driven by the terminal end buds (TEBs) structures located at the tips of the ducts, and through a process of secondary side branching. Once the ducts reach the edges of the mammary fat pad, the TEBs regress and the arborized ductal tree remains quiescent until pregnancy, except for the formation of tertiary side branches at each oestrus cycle[1]. During pregnancy a massive process of tertiary side branching results into lobuloalveolar structures that ultimately differentiate into alveoli, producing and secreting milk during lactation[2]. At weaning, the MG reverts to the adult virgin-like MG state. During each phase of MG development profound changes in cell proliferation, apoptosis and differentiation occur, allowing a constant remodelling of the gland structure.

The MG is composed of two compartments, the epithelial ducts and the stroma in which ducts are embedded. Postnatal MG development is controlled globally by systemic hormones and locally by the repetitive and bi-directional cross-talk between the epithelial and the stromal compartment[3]. Many studies have shown that mammary stroma instructs differentiation and branching of mammary epithelial cells through paracrine signals, and sustains cell fate maintenance during postnatal life[4]. The paracrine signalling between epithelial and stromal cells is based on secreted factors like the Wnt ligands[5,6].

The Wnt signalling pathway control multiple steps of MG development and dysregulation of Wnt signalling has been associated with breast cancer (BC)[7]. While mutations in the core genes of the canonical Wnt signalling pathway, namely *βcatenin*, *APC*, or *axin*2, are rare, loss by epigenetic silencing of Wnt inhibitors like secreted frizzled-related protein (Sfrp) family members is more frequent, and it has been associated with human BC[8].

The Sfrps are a family of secreted glycoproteins composed of five members in mammals, containing an N-terminal domain homologous to the cysteine-rich domain (CRD) of the Frizzled Receptor and a C-terminal domain with some homology to the netrin domain (NTR)[9]. Sfrp3 (also known as FRZB) is considered as the founding member of the Sfrp family[10], and identified as an inhibitor of Wnt signalling[11,12]. Sfrp3 has an essential role in bone development and cartilage homoeostasis, dependent and independent of Wnt signalling[13–15]. Remarkably, we identify Sfrp3 as a potential regulator of 3D-epithelial polarity[16]. Indeed, Sfrp3 levels are downregulated in different human epithelial cancers, underscoring its importance for normal epithelial architecture[17]. Recent reports demonstrated that human BC lack expression of *Sfrp3*[16], and also a positive association between polymorphisms of *Sfrp3* and predisposition to BC[18]. However, whether Sfrp3 has a fundamental role in MG morphogenesis and cancer onset was previously unknown.

In the present study, we show that loss of *Sfrp3* results in increased ductal elongation and tertiary branching during pubertal MG development. This advanced lobuloalveolar development is accompanied by structural defects of epithelial ducts. *Sfrp3* knockout luminal epithelial cells show increased proliferation, with loss of polarity features, and they are abnormally differentiated into milk-secreting cells. Transplantation experiments and 3D cultures reveal that the ductal phenotype results from the loss of stromal *Sfrp3*. Also, we demonstrate that SFRP3 functions as an extracellular transporter of Wnts using the larval model of *Drosophila*, and show that SFRP3 control the levels of Wnts that accumulate in the stromal–epithelial boundary, thus modulating the proliferative and branching response of the epithelium during MG morphogenesis. Finally, we identify that lack of Sfrp3 in

MMTV-Neu animals induces the precocious appearance of premalignant lesions, suggesting a role for Sfrp3 as a marker for BC susceptibility. Collectively, our results show that stromal *Sfrp3* regulates the Wnt-mediated epithelial–stromal interactions in the MG development.

## Results

**Stromal and epithelial cells of the MG express Sfrp3**. In order to investigate the role of *Sfrp3* in MG morphogenesis, we examined the expression of *Sfrp3* during MG development. *Sfrp3* mRNA expression is high at the pre-pubertal stage (3 weeks old) and decreases throughout puberty (6–8 weeks old). During pregnancy, its expression is further reduced (Fig. 1a). We also examined the expression of all *Sfrp* family members during MG development. Similarly, *Sfrp2, 4* and *five* mRNA expression are high at the pre-pubertal stage and decreases throughout puberty, whereas, the levels of sfrp1 increases (Supplementary Fig. 1a). To determine which cellular compartment within the MG expresses *Sfrp3*, we separated stromal and epithelial cells from adult MGs (12 weeks old), based on their expression of CD49f and EpCAM cell surface markers[19] (Supplementary Fig. 1b). *Sfrp3* is highly expressed in stromal cells compared to relatively low levels present in the epithelial compartment, which includes basal/myoepithelial and luminal cells (Fig. 1c). Furthermore, we detected Sfrp3 mRNA in the mammary cleared fat pad, showing a decline during puberty (Fig. 1b). To better dissect the source of Sfrp3, we isolated the most abundant cell populations of the stroma, and we identified that Sfrp3 is mostly expressed by fibroblasts (Supplementary Fig. 1c).

In summary, we found that *Sfrp3* is highly expressed at prepuberty before mammary ducts morphogenesis occurs, and it is more abundant in the stromal compartment. The developmentally regulated pattern of expression of *Sfrp3* suggests a potential role of *Sfrp3* on MG development.

**Sfrp3 prevents early ductal invasion and side branching**. To address the function of *Sfrp3* during MG development, we used the *Sfrp3* knockout mouse[14]. Homozygous (*Sfrp3*$^{-/-}$) mice are viable with no apparent morphological postnatal phenotype[14]. We confirmed that *Sfrp3* expression is absent in *Sfrp3*$^{-/-}$ mice in MGs (Supplementary Fig. 1d, e). To detect defects in MG morphology, we examined wild-type and *Sfrp3*$^{-/-}$ virgin females with matching oestrous cycles at different developmental stages. Both wild-type and *Sfrp3*$^{-/-}$ mice present a regular mammary epithelial tree in pre-puberty (Supplementary Fig. 1f). Differences start to be evident at mid-puberty, when epithelial ducts are markedly longer in *Sfrp3*$^{-/-}$ MGs, almost reaching the edges of the fat pad (Fig. 1d, e). At the end of puberty, *Sfrp3*$^{-/-}$ ducts not only show an accelerated invasion into the fat pad, but also a four-fold increase increased side branching (Fig. 1d, e). *Sfrp3*$^{-/-}$ branches terminate with small protrusions, lobuloalveolar buds, which are typical of pregnancy (Fig. 1f). This precocious hyperbranched phenotype persists in adult *Sfrp3*$^{-/-}$ (Fig. 1d, e). These results suggest that *Sfrp3* controls MG development by preventing unwarranted mammary epithelial invasion and branching during puberty.

**Sfrp3 prevents ducts defects and altered differentiation**. To determine whether structural defects in *Sfrp3*$^{-/-}$ epithelial ducts, we histologically analysed MG. *Sfrp3*$^{-/-}$ MG sections confirmed the hyperbranched phenotype and further revealed a lumen filling phenotype of the ducts (Fig. 2a). Whereas control ductal structures are characterised by a normal hollowed lumen, in *Sfrp3*$^{-/-}$ MGs the lumen of 30% ducts is partially or entirely occluded (Fig. 2a, b). *Sfrp3*$^{-/-}$ ducts display remarkable alterations in the

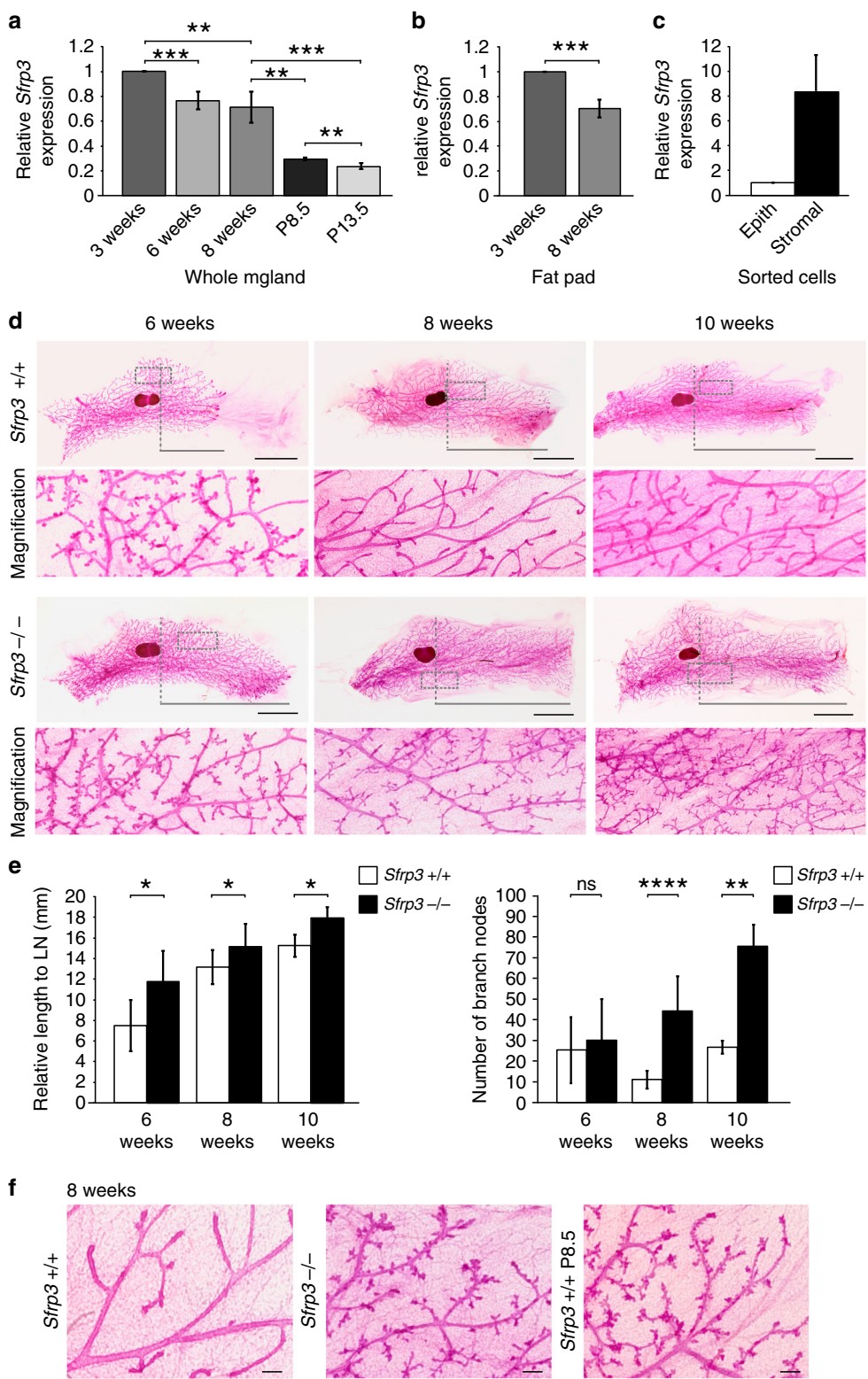

structure of the luminal compartment, while the basal compartment is normal (Fig. 2c). The inner luminal epithelium of *Sfrp3*$^{-/-}$ ducts is abnormally organised into multiple layers, filling in the luminal space of ducts and branches, compared to the single inner layer of luminal cells of the controls (Fig. 2c, d). Next, we analysed the distribution of polarity markers. In wild-type ducts, E-cadherin and ZO1 are localised at adherens and tight junctions, respectively, of luminal cells (Fig. 2e). In *Sfrp3*$^{-/-}$ ducts only the most apically localised luminal cells express

E-cadherin and ZO1, which have both a normal distribution, comparable to controls. The expression of ZO1 is absent in the most internal cells, whereas E-cadherin distributes uniformly and widely along the entire cell membrane of *Sfrp3*$^{-/-}$ ducts (Fig. 2e).

Next, we determined whether the multistratified arrangement of *Sfrp3*$^{-/-}$ luminal cells is due to either an increase in proliferation rate or a decrease in apoptosis rate. In control MG only a few cells are proliferating, and almost no apoptotic cells are

**Fig. 1** Epithelial invasion and tertiary-side branching increase in *Sfrp3*−/− MG. **a** Relative *Sfrp3* mRNA level in whole MG at 3, 6, 8 weeks (weeks) of age and at pregnancy (P) day 8.5 and 13.5. *n* = 4 females/age, except for P8.5, *n* = 3 females. **b** Relative *Sfrp3* mRNA level in cleared mammary fat pad of females at 3 and 8 weeks of age. *n* = 4 females/age. **c** Relative *Sfrp3* mRNA level in FACS-sorted mammary epithelial and stromal cells. *n* = 2 pools, each pool collected from 15 females at 12 weeks of age. **d** Representative whole mount of MGs from *Sfrp3*+/+ and *Sfrp3*−/− virgin females at 6, 8 and 10 weeks of age. The area bracketed in each whole mount image is shown below at higher magnification. Grey horizontal line illustrates the distance from the lymph node (dashed vertical line) to the longest duct. Scale bar: 0.5 cm. **e** Quantification of the ductal length relative to the lymph node (LN) (left) and quantification of the number of branch nodes (right) in MGs from *Sfrp3*+/+ and *Sfrp3*−/− virgin females at 6 (*n* = 4), 8 (*n* = 9), and 10 weeks (*n* = 2 *Sfrp3*+/+; 4 *Sfrp3*−/−) of age. **f** Representative whole mount of MGs from *Sfrp3*+/+ and *Sfrp3*−/− virgin females at 8 weeks of age compared to *Sfrp3*+/+ pregnant females (P8.5) of the same age. Scale bar: 0.25 cm. For **a–c**, **e**, data represent mean ± SD (*$p < 0.05$, **$p < 0.01$, ***$p < 0.001$, ****$p < 0.0001$, ns, not significant). See also Supplementary Fig. 1. Source data are provided as a Source Data file

detected (Fig. 3a, b), which is consistent with the entering of post-pubertal virgin females into a quiescent stage until pregnancy. There is a dramatic increase in the level of proliferation in *Sfrp3*−/− mature luminal cells, whereas the number of apoptotic cells is similar to control MGs (Fig. 3a–c). Accordingly, gene expression profiling of mammary epithelial cells from adult *Sfrp3*−/− mice showed an mRNA increase of classical cell cycle markers such as *Ccnd2*, *Klf5*, *Fhl2*, *Id4* (Supplementary Data 1). Luminal cells are also classified by their expression of hormone receptors, particularly oestrogen receptor (ER). While ducts contain both ER− and ER+ luminal cells, alveolar luminal cells are mainly ER−[20,21]. We identified a substantial decrease in the ER+ population in the *Sfrp3*−/− MGs (Fig. 3d, e), that suggest a precocious alveolar development of *Sfrp3*−/− MGs.

The phenotype of *Sfrp3*−/− MGs resembles the state of early pregnancy in wild-type mice (P8.5) when epithelial cells undergo a process of proliferation and differentiation to form acini-like structures that ultimately mature into alveoli—the milk secretory structures—during pregnancy and lactation. This similarity, together with the critical increase in the number of ER− cells, suggested an altered differentiation of *Sfrp3*−/− MGs. Remarkably, MGs of adult virgin *Sfrp3*−/− virgin females expressed high levels of ß casein compared to controls (Fig. 3f, g). ß casein was detected inside the lumen and within luminal cells with a subapical localisation in *Sfrp3*−/− mammary ducts (Fig. 3f). Consistently, we detected higher levels of other two milk proteins, whey acidic protein (WAP) and alpha-lactalbumin (Lalba), in virgin *Sfrp3*−/− glands (Fig. 3g). Further, the expression of the lactating alveolar marker sodium–inorganic phosphate (NaPi)-type IIb is substantially increased (Fig. 3h). This premature differentiation does not impair the capability of *Sfrp3*−/− MGs to undergo the normal morphological changes associated with pregnancy (Supplementary Fig. 2a, b), suggesting that alveoli are functional and milk can flow through ducts, despite the occlusion of the luminal space in *Sfrp3*−/− MGs.

We further investigated whether *Sfrp3*−/− showed alterations in the stromal compartment. While we found some differences in the extracellular matrix organisation between wild-type and *Sfrp3*−/− MGs, they were not statistically relevant (Supplementary Fig. 2c–f).

Together, these observations suggest that loss of *Sfrp3* leads to a disruption of the mammary ductal structures in virgin females, with overproliferating, and weakly polarised ER− luminal epithelial cells that arrange in multiple layers facing the ductal lumen and that are abnormally differentiated to a pregnancy state.

**Sfrp3 is expressed by the stroma to control MG development.** Since both stromal and epithelial cell compartments expressed Sfrp3 mRNA (Fig. 1c), we wondered whether the branching phenotype of *Sfrp3*−/− mice is due to loss of *Sfrp3* either in the stroma or in the epithelium. To address this issue, we performed mammary epithelial transplantation experiments, consisting in the transplantation of mammary epithelial tissue from a donor

adult mouse into the cleared fat pad of a recipient prepubertal mouse. Five weeks after transplantation we analysed the transplanted and the contralateral MGs (Supplementary Fig. 3a, b).

As expected, wild-type mammary tissue, when transplanted into wild-type cleared fat pad, grew and developed into an epithelial tree showing a regular branching pattern, while *Sfrp3*−/− transplanted epithelial cells into an *Sfrp3*−/− cleared fat pad formed hyperbranched epithelial ducts (Supplementary Fig. 3c). When *Sfrp3*−/− mammary epithelial cells were transplanted into wild-type fat pad, they grew and invaded the stroma, giving rise to a regular epithelial ductal tree similar to that observed in the contralateral MG (Fig. 4a, b). We obtained similar results with immunocompromised females as transplant recipients (Supplementary Fig. 3b, d). In contrast, wild-type tissue transplanted into *Sfrp3*−/− stroma generated almost four-fold increase hyperbranched epithelial network, comparable to that observed in the *Sfrp3*−/− MGs (Fig. 4a–c). We obtained a similar phenotype when wild-type tissue was transplanted into *Sfrp3*−/− cleared fat pad, and allocated in immunocompromised females, which discards a systemic effect of Sfrp3 (Supplementary Fig. 3f). These results suggest that loss of *Sfrp3* in the stroma is likely the cause of *Sfrp3*−/− phenotype in MGs. Indeed, wild-type outgrowths transplanted in *Sfrp3*−/− stroma show a phenotype similar to *Sfrp3*−/− MGs, with multilayering of luminal epithelial cells, polarity defects, and precocious alveolar differentiation (Fig. 4d, e), which suggests that loss of stromal *Sfrp3* leads to loss of ductal architecture. In contrast, *Sfrp3*−/− outgrowths transplanted in wild-type fat pad form a network of ducts with a proper architecture (Fig. 4d, e, Supplementary Fig. 3e)

Consistently with the transplantation results, we did not detect any substantial difference between wild-type and *Sfrp3*−/− mammary epithelial organoids in terms of either organoid structure/size or lumen volume. Also, in the absence of growth factors neither wild-type nor *Sfrp3*−/− organoids form branches (Supplementary Fig. 4a). Neither, we did observe any substantial difference in the frequency of wild-type and *Sfrp3*−/− epithelial cells able to generate mammospheres, which is used to estimate the number of mammary stem cells (MaSC) in the epithelial tissue[22] (Supplementary Fig. 4a). Together, these results indicate that the abnormal features of the ductal phenotype in *Sfrp3*−/− MG arise from loss of *Sfrp3* in the stroma, rather than in the epithelium.

**Sfrp3 functions in a hormonally stimulated environment.** Ovarian oestrogen and progesterone are the central systemic regulators of MG development during puberty and pregnancy. Higher levels of progesterone[23] accompany the formation of alveolar-like structures during the diestrus phase of the oestrous cycle and pregnancy[23]. In the absence of ovarian hormones, *Sfrp3*−/− ducts do not elongate and do not show any alteration in branching rate or structural organisation (Supplementary Fig. 4b). Additionally, we detected no increase in the serum levels of oestrogen and progesterone in *Sfrp3*−/− females

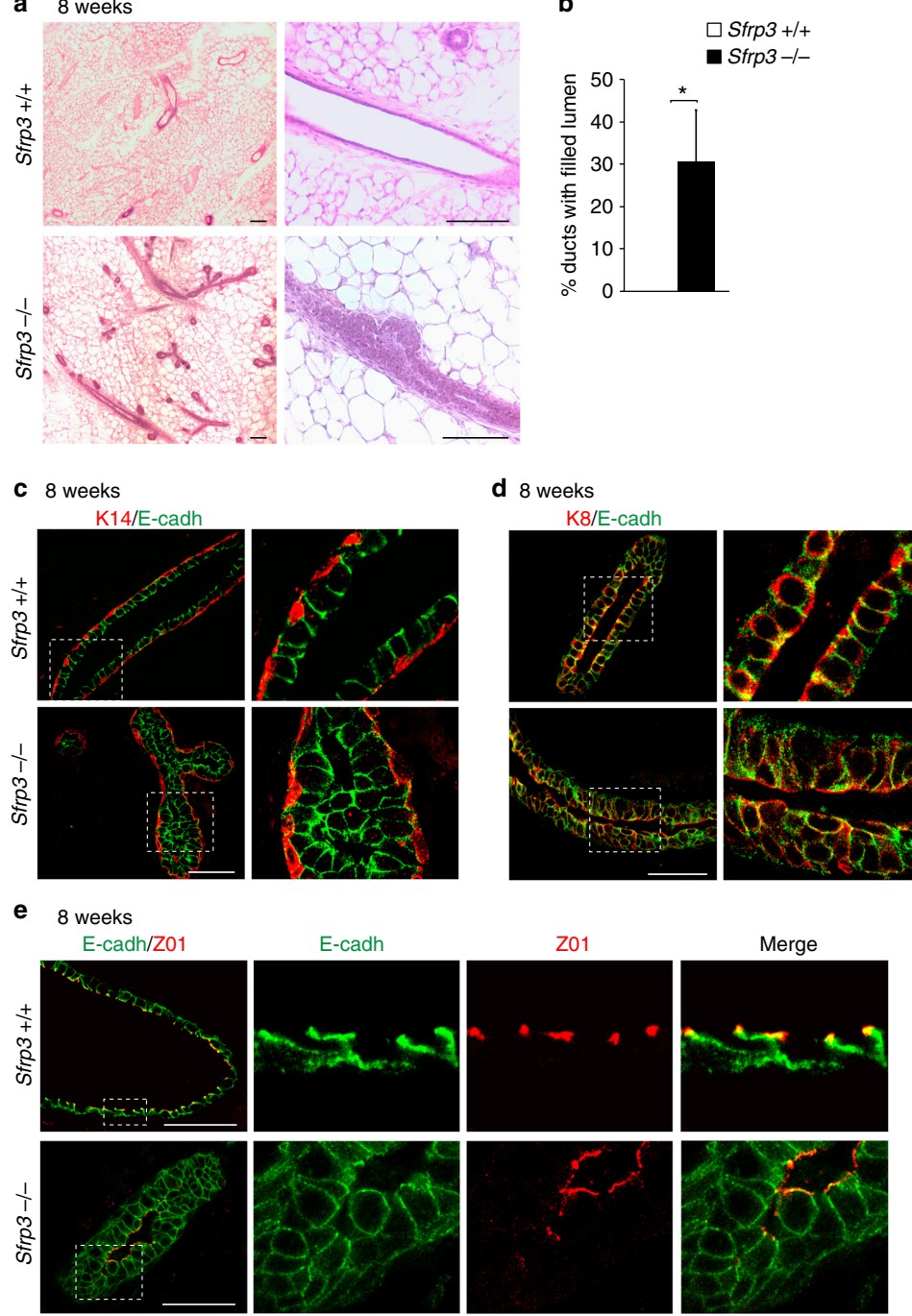

**Fig. 2** *Sfrp3* loss disrupts lumen formation and epithelial architecture of ducts. **a** Representative images of HE-stained mammary glands from *Sfrp3*^+/+ and *Sfrp3*^−/− virgin females at 8 weeks of age. Scale bar: 100 μm. **b** Quantitative analysis of the luminal filling phenotype showing an increase in the number of ducts with occluded lumen in *Sfrp3*^−/− compared to *Sfrp3*^+/+ females at 8 weeks of age. $n = 2$ *Sfrp3*^+/+; 3 *Sfrp3*^−/−. Data represent mean ± SD (*$p < 0.05$). **c**, **d** Immunofluorescence staining for Keratin 14 (K14) and E-cadherin (E-cadh) (**c**) and for Keratin8 (K8) and E-cadherin (**d**) on mammary sections of 8 weeks old females. The magnification of each bracketed area is shown on right side. Scale bar: 50 μm. **e** Immunofluorescence staining for E-cadherin and ZO1 on mammary sections of 8 weeks old females. **c–e** Images are representative of at least 10 MG sections analysed in 5 different experiments. Scale bar: 50 μm. See also Supplementary Fig. 2. Source data are provided as a Source Data file

(Supplementary Fig. 4c), precluding that the branching phenotype of *Sfrp3*^−/− mice is due to an increase in hormone levels. Indeed, in *Sfrp3*^−/− males (complete absence of ovarian function) did not show any induction of the mammary tree (Supplementary Fig. 4d). In summary, these results indicate that stromal *Sfrp3* modulated epithelial MG development only in a hormonally stimulated environment.

**Sfrp3 modulate of Wnt signalling in MG development**. Taking into account that Wnt signalling pathway has been traditionally associated with MG development[7], and Sfrps are regulators of Wnt signalling[11,12], we hypothesise that Sfrp3 might control mammary development through modulating Wnt signalling. However, Sfrps present roles that could be either dependent or independent of Wnt[13–15].

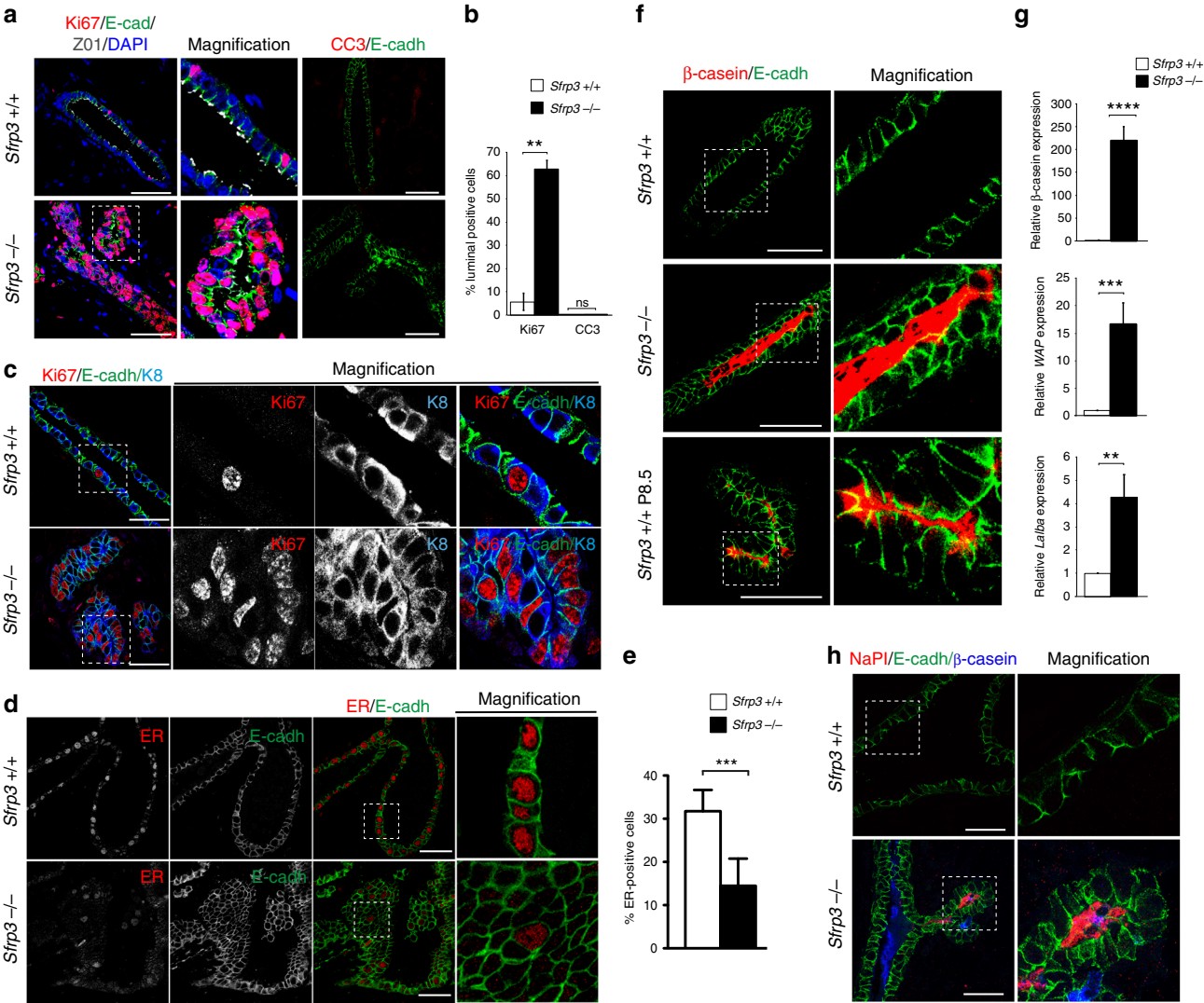

**Fig. 3** Increased proliferation and premature differentiation in *Sfrp3*$^{-/-}$ MG. **a** Representative images of immunofluorescence staining for Ki67, E-cadherin, ZO1 and DAPI (left and middle) and for Cleaved Caspase 3 (CC3) and E-cadherin (right) on 8 weeks old MGs. Scale bar: 50 μm. **b** Quantification of the number of Ki67 and CC3 positive cells in mammary ducts of 8 weeks old *Sfrp3*$^{+/+}$ (*n* = 3) and *Sfrp3*$^{-/-}$ (*n* = 2) females. Data represent mean ± SD (**p < 0.01). **c** Representative images of immunofluorescence staining for Ki67, E-cadherin, Keratin 8 on 8 weeks old mammary glands. Scale bar: 50 μm. **d** Representative images of immunofluorescence staining for ER (red) and E-cadherin (green), on 8 weeks old mammary glands. Scale bar: 50 μm. **e** Quantification of the number of ER positive cells in mammary ducts of 8 weeks old *Sfrp3*$^{+/+}$ (*n* = 2) and *Sfrp3*$^{-/-}$ (*n* = 2) females. Data represent mean ± SD (***p < 0.001). **f** Representative images of immunofluorescence staining for ß casein and E-cadherin on mammary section from 10 weeks old *Sfrp3*$^{+/+}$ and *Sfrp3*$^{-/-}$ virgin females and *Sfrp3*$^{+/+}$ pregnant females (P8.5). Scale bar: 50 μm. **g** Relative mRNA level of ß casein (upper), WAP (middle) and *Lalba* (bottom) in MGs of *Sfrp3*$^{+/+}$ (*n* = 3) and *Sfrp3*$^{-/-}$ (*n* = 4) virgin females at 10 weeks of age. **h** Representative images of immunofluorescence staining for NaPI (red), E-cadherin (green), ß casein (blue) on 8 weeks old MGs. **c**, **h** Images are representative of at least 5 MG sections analysed in three different experiments. Data represent mean ± SD (**p < 0.01, ***p < 0.001, ****p < 0.0001). See also Supplementary Fig. 2. Source data are provided as a Source Data file

To address the role of *Sfrp3* we used the wing imaginal disc of *Drosophila*, a well-studied model to investigate morphogens during development. *Drosophila* represents a natural *Sfrp3* null background since no *Sfrp* homologue has been identified in its genome. We observed that ectopic expression of Sfrp3 in the posterior compartment of the *Drosophila* wing imaginal disc (driven by HhGal4) does not interfere with the symmetrical gradient of Wingless (Wg) at the dorsoventral boundary of the wing imaginal disc, but reduced its peak of concentration there (Fig. 5a, b). Sfrp3 expression also dramatically abolishes the expression of Senseless, a canonical target of Wg, but not the expression of Cut, a target of Notch signalling (Fig. 5c, d). Consistently, the adult wings of *Drosophila* expressing

NubGal4>UAS-Sfrp3 show a lack of sensory organs at the wing margins, which is a characteristic phenotype of the inhibition of Wg pathway (Fig. 5c). These results suggest that Sfrp3 acts as an inhibitor of Wg pathway in *Drosophila* wing-imaginal-disc. To validate this hypothesis, we also characterised the effect of Sfrp3 on Distalless (dll), a Wg target that response to low Wg concentrations. We observed that Sfrp3 expression does not inhibit dll, but instead broadens the dll territory in the P compartment (Fig. 5f). Thus, while the effect on Sen seems to support the conclusion that Sfrp3 overexpression inhibits Wg, the effect on dll suggests that Sfrp3 expression might induce the transport and diffusion of Wg, reducing its concentration at the D/V boundary. This change in Wg would inhibit Sens

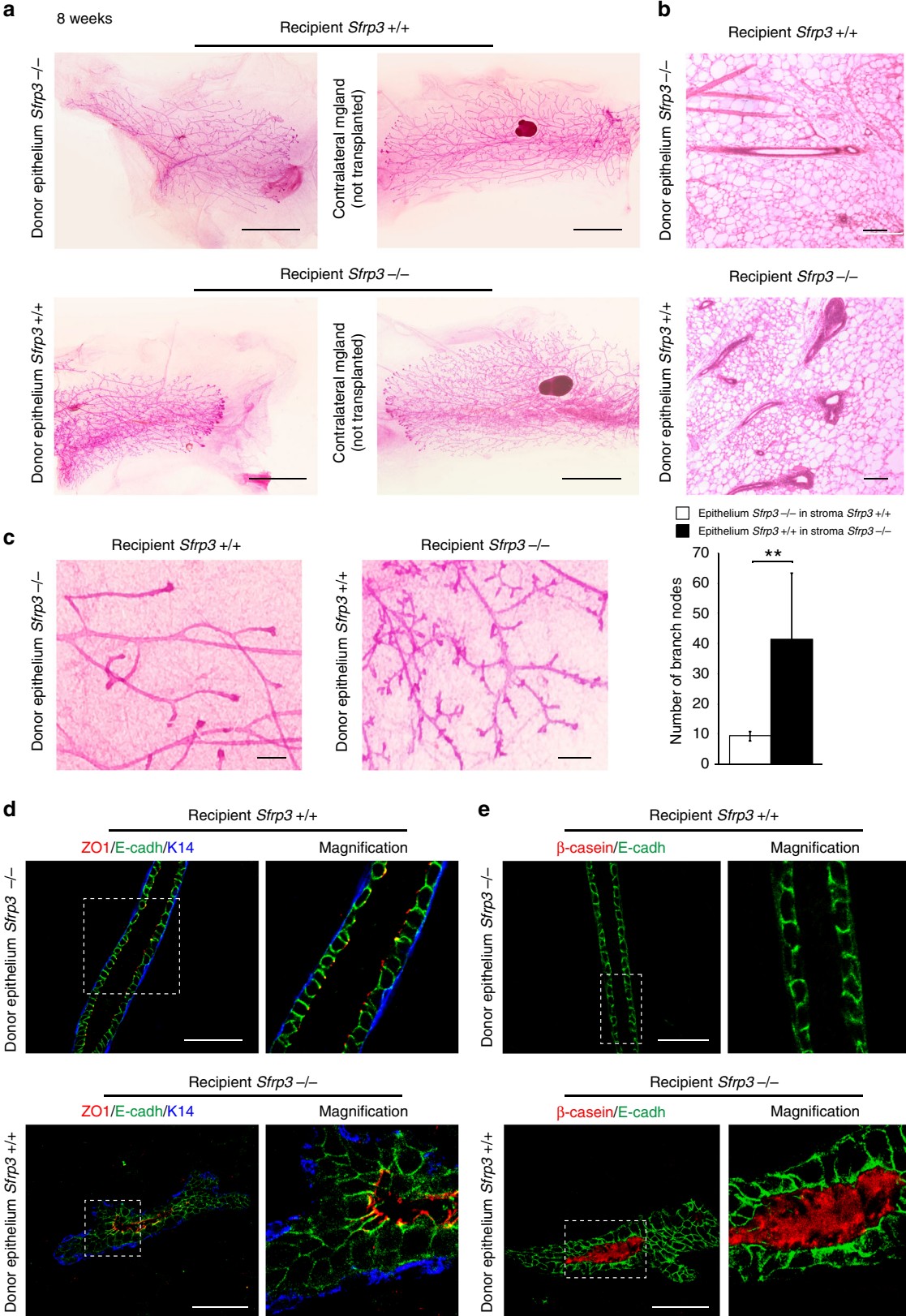

(a high-concentration Wg-target) but expand dll territory (a low-concentration Wg-target). In order to prove this role of Sfrp3 in the transport and diffusion of Wg, we used a membrane-bound form of Wg (NRT-Wg), described before to allow persistent expression of relevant target genes[24]. Importantly, the expression of Sfrp3 in the posterior compartment did not affect Sens

(Fig. 5g), demonstrating that Sfrp3 is not a repressor or inhibitor of Wg, but more likely a transporter of Wg that allow the spreading of this morphogen.

Next, we investigated whether canonical Wnt signalling is abnormally activated in $Sfrp3^{-/-}$ MGs. Next-generation sequencing (NGS) analysis of sorted adult MG epithelial cells showed a

**Fig. 4** Loss of stromal *Sfrp3* affects branching, ductal structure and differentiation. **a** Whole mount of transplanted and contralateral (not transplanted) MG of CD1 females analysed 5 weeks after the operation. Upper panel, left, *Sfrp3*$^{-/-}$ donor epithelium is transplanted in *Sfrp3*$^{+/+}$ stroma; bottom panel, left, *Sfrp3*$^{+/+}$ donor epithelium is transplanted in *Sfrp3*$^{-/-}$ stroma. Images are representative of $n = 7$ transplants of *Sfrp3*$^{-/-}$ donor epithelium in *Sfrp3*$^{+/+}$ stroma; 7 transplants of *Sfrp3*$^{+/+}$ donor epithelium in *Sfrp3*$^{-/-}$ stroma. Scale bar: 0.5 cm. **b** HE-stained transplanted MGS of CD1 females. Scale bar: 50 μm. **c** Left, high magnification view of transplanted MGs. Scale bar: 0.3 cm. Right, quantification of the number of branch nodes in *Sfrp3*$^{+/+}$ MGs transplanted with *Sfrp3*$^{-/-}$ donor epithelium ($n = 5$) and in *Sfrp3*$^{-/-}$ MGs transplanted with *Sfrp3*$^{+/+}$ donor epithelium ($n = 4$). Data represent mean ± SD (\*\*$p < 0.01$). **d** Images of immunofluorescence staining for ZO1 (red), E-cadherin(green) and K14 (blue) in *Sfrp3*$^{+/+}$ MGs transplanted with *Sfrp3*$^{-/-}$ donor epithelium (upper panel) and in *Sfrp3*$^{-/-}$ MGs transplanted with *Sfrp3*$^{+/+}$ donor epithelium (bottom panel) ($n = 7$). Scale bar: 50 μm. **e** Images of immunofluorescence staining for ß casein (red) and E-cadherin (green) in *Sfrp3*$^{+/+}$ MGs transplanted with *Sfrp3*$^{-/-}$ donor epithelium (upper panel) and in *Sfrp3*$^{-/-}$ MGs transplanted with *Sfrp3*$^{+/+}$ donor epithelium (bottom panel) ($n = 7$). **d**, **e** Images are representative of at least 10 MG sections analysed. Scale bar: 50 μm. For **d** and **e** the magnification of the bracketed area is shown on right side. See also Supplementary Figs. 3 and 4. Source data are provided as a Source Data file

substantial up-regulation of *LEF1* and *cMYC* in *Sfrp3*$^{-/-}$ epithelial cells (Supplementary Data 1). However, the expression levels of most canonical target genes like *βcatenin*, *axin2*, and *cyclinD1* are unchanged in *Sfrp3*$^{-/-}$ adults (Supplementary Fig. 5, Supplementary Data 1). By contrast, we observed a substantial nuclear accumulation of βcatenin in pubertal *Sfrp3*$^{-/-}$ MGs (Fig. 6a, b), when the levels of expression of Sfrp3 are close to the maximum level (Fig. 1a). Together these results demonstrate that Sfrp3 acts as a specific modulator of canonical Wnt pathway.

Previous studies have shown that Wnt family members are differentially regulated in mouse MGs during puberty and pregnancy[6]. Since *Sfrp3*$^{-/-}$ MGs mimics an early pregnancy state, we investigated whether the loss of *Sfrp3* associates with an alteration of Wnt ligands. We defined the cellular compartments that express these Wnt ligands by cell sorting and qPCR analyses. We determined that *Wnt2* is more abundant in the stroma, while *Wnt5a* and *Wnt7b* are more expressed in the epithelial cells in adult virgin MGs (Fig. 6c). In line with this, NGS analysis detected shallow levels of *Wnt2* and higher levels of *Wnt5a* and *Wnt7b* in wild-type epithelial cells (Supplementary Data 1). Furthermore, we found that Wnt2 protein accumulated at the stromal–epithelial boundary in *Sfrp3*$^{-/-}$ (Fig. 6d), while de-epithelialised cleared fat pads of adult animals presented reduced levels of expression of *Wnt2* (Fig. 6e), which is consistent with the decrease we observed in pregnant mice for *Wnt2* (Supplementary Fig. 6). This data suggests that *Sfrp3* function might be related to control the accumulation of Wnt2 at the stromal–epithelial boundary since the loss of stromal *Sfrp3* leads to an accumulation of *Wnt2* in the boundary, which is consistent with the role found for Sfrp3 in *Drosophila* before. Altogether, our results suggest that stromal *Sfrp3* mediates the crosstalk between stroma and epithelium through the modulation of stromal *Wnt2* ligand.

To obtain a more precise idea of the molecular mechanism of Sfrp3, we studied the secretion of SFRP3 in vitro and its interaction with Wnt ligands. We found that myc-tagged SFRP3 requires the NTR domain to be secreted (Fig. 6f). Next, we analysed the interaction of SFRP3 with different Wnts present in the MG[6]. SFRP3 binds strongly to Wnt2 that is expressed by the stromal cells, and to a lesser extent with Wnt5a or Wnt7b expressed by the epithelial cells (Fig. 6g). Furthermore, we observed that the NRT domain of SFRP3 binds to all Wnts analysed, but this interaction is particularly strong with Wnt2 (Fig. 6h). The specific interaction of the NRT domain of Sfrp with Wnts/Wg was previously described for SFRP1[25,26], another member of the family. These results, combined with our previous data, suggest that SFRP3 interacts with Wnt2 to control the concentration levels of Wnt2 in the boundary of the stromal–epithelial mammary.

**Loss of *Sfrp3* associates with cancer.** *Sfrp3*$^{-/-}$ phenotype might resemble early stages of carcinogenesis[27,28]. However, we could

not detect any defined tumour lesion like focal hyperplasia, disruption of the myoepithelial layer or change in collagen content in the *Sfrp3*$^{-/-}$ mice. Neither, did we observe any palpable tumour mass, or macroscopic lesion in nulliparous *Sfrp3*$^{-/-}$ females of 12/15 month of age. We also monitored *Sfrp3*$^{-/-}$ females mated and allowed to undergo several rounds of pregnancy, lactation, and involution. Again, *Sfrp3*$^{-/-}$ mice did not develop any mammary tumour lesion, indicating that aging and parity are not sufficient risk factors for tumour development in *Sfrp3*$^{-/-}$ mice. Although these results suggest that, in the absence of genetic changes that could provide a sensitised background, *Sfrp3* does not act as tumour suppressor, we performed an NGS analysis of sorted mammary gland epithelial cells to identify transcriptomic changes pointing to tumorigenic susceptibility.

A total of 1408 differentially expressed genes (DEGs, *q*-value < 0.05) were identified, including 813 upregulated and 595 down-regulated genes in *Sfrp3*$^{-/-}$ compared to wild-type mammary epithelial cells (Supplementary Data 1). From the top 100 DEGs (50 upregulated and 50 downregulated) (Supplementary Data 2, Supplementary Data 3, Fig. 7a) we selected a group of 56 genes with a log2 fold change >2 or <−2. To investigate their association with BC outcomes, we interrogated two independent breast tumour datasets deposited in the publicly available Oncomine database[29], and we found data for 36 genes. Importantly, 15 upregulated genes in *Sfrp3*$^{-/-}$ epithelial cells are differentially expressed in BC versus normal samples. We observed that the expression of 8 genes (*Slc1a2*, *Larp1b*, *Lef1*, *Edn2*, *Slc6a12*, *Fgf21*, *C1qtnf5* and *Flt3l*) is appreciably elevated in invasive BC compared to normal breast samples using the Cancer Genome Atlas (TCGA) dataset (Fig. 7b). All eight genes are highly expressed in invasive ductal and/or lobular BC, except for *Flt3l* that is notably highly expressed only in invasive lobular BC, compared to controls. Furthermore, seven genes (*Ecel1*, *Nr4a2*, *Plk3*, *Dpp10*, *Saa2*, *Csn2* and *Nrxn3*) are considerably highly expressed in the stroma of invasive BC (Fig. 7c).

We found similar evidence when we performed a tissue-specific analysis of NGS data. From the total 1408 DEGs, we selected 102 DEGs (74 upregulated and 28 downregulated genes) that have been already described in the literature as associated with mammary gland for expression or functional data (Supplementary data 4) and we classified them into 20 KEGG pathways. Among the top four false discovery rate (FDR) ≤ 10%, we found "Proteoglycans in cancer" "and "Pathways in cancer" signalling pathways (Fig. 7d). These data show that loss of *Sfrp3* results in an epithelial gene expression profile that reflects oncogenic processes. Finally, when classified into the biological process (BP), the clusters with the top three enrichment scores include processes of regulation of cell proliferation, positive regulation of protein metabolic process and mammary gland alveolus development (Supplementary Table 1).

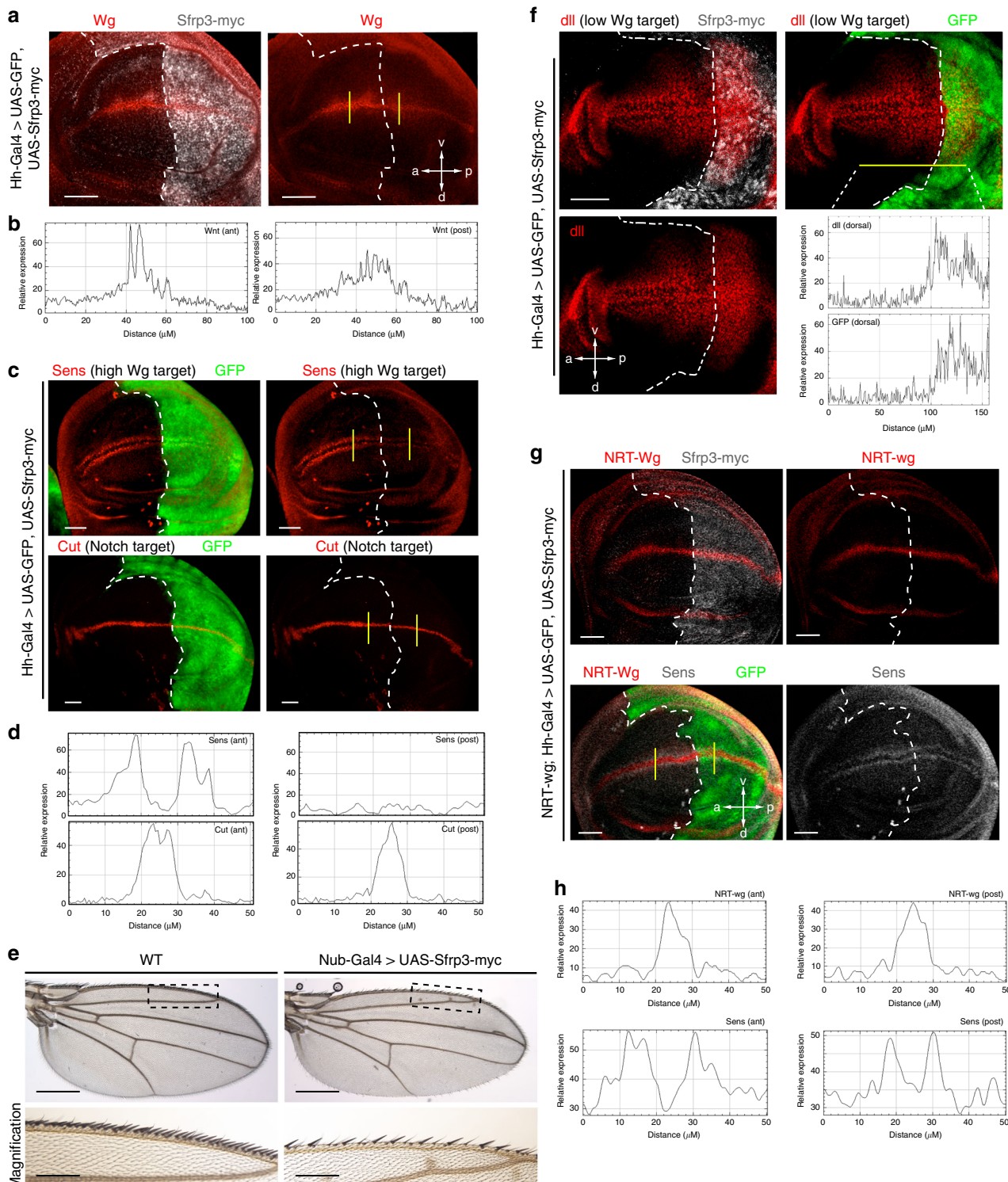

To further test the potential of sfrp3 as tumour suppressor in the presence of genetic changes that provide a sensitised background, we performed a crossed $Sfrp3^{-/-}$ mice with the MMTV-Neu (MMTV-Erbb2), which produced tumours classified as human luminal type carcinomas[30]. We observed the formation of preneoplastic lesions in the sfrp3$^{-/-}$ MMTV-Neu mammary gland samples which are phenotypically similar to the MMTV-Neu premalignant lesions described before (Fig. 7e)[30]. Furthermore, we identified that lack of Sfrp3 in MMTV-Neu animals induces a significative reduction in the time of appearance of mammary tumours (from an estimated median of 405 days in Srrp3$^{+/-}$ animals to 344 days in Sfrp3$^{-/-}$ mice; $p = 0.0148$) (Fig. 7f), suggesting a role for Sfrp3 as relevant marker for BC susceptibility. The analysis of the Sfrp3$^{-/-}$; MMTV-Neu mammary gland tumours by confocal microscopy showed that the tissues present the characteristic features of the Sfrp3$^{-/-}$ mammary ducts with β-casein accumulated in the lumen of ducts and hyperproliferative luminal cells (Fig. 7g).

Collectively, these results support the alterations in MG morphogenesis we observed in $Sfrp3^{-/-}$ virgin females, and reveal Sfrp3 as a potential marker for BC susceptibility.

**Fig. 5** *Sfrp3* modulates Wnt signalling in the *Drosophila* wing imaginal disc. **a** Wingless (Wg) expression is altered in wing imaginal disc expressing Sfrp3-myc (grey) in the posterior compartment under Hh-Gal4 driver compared to the anterior compartment, which serves as control. Scale bar: 50 μm. **b** Fluorescence intensity profile of Wg (grey value; arbitrary units) along yellow lines in the anterior (ant) and posterior (post) compartment of the wing imaginal disc. **c** In Hh-Gal4>UAS-GFP, UAS-Sfrp3-myc imaginal discs, the Wingless target Senseless (Sens) expression is inhibited only in the posterior compartment (delimited by GFP expression) but not in the anterior compartment that serves as control. On the contrary, the Notch target Cut is normally expressed in these imaginal discs along the anterior-posterior axis. Scale bar: 50 μm. **d** Fluorescence intensity profile of Sens and Cut (grey value; arbitrary units) along yellow lines in the anterior (ant) and posterior (post) compartment of the wing imaginal disc. **e** Adult Nub-Gal4>UAS-Sfrp3-myc wings expressing Sfrp3 in the entire wing pouch and hinge show a lack of sensory organs along the margins that correspond to the dorsoventral boundary in imaginal discs. A high magnification of the bracketed area is shown. **f** In Hh-Gal4>UAS-GFP, UAS-Sfrp3-myc imaginal discs, the Wingless target Distalless (dll) expression is expanded in the posterior compartment (delimited by GFP/Sfrp3 expression) but not in the anterior compartment that serves as control. Scale bar: 50 μm. Fluorescence intensity profile of dll and GFP (grey value; arbitrary units) along the dorsal yellow lines of the wing imaginal disc. **g** In NRT-wg; Hh-Gal4>UAS-GFP, UAS-Sfrp3-myc imaginal discs, the Wingless target Senseless (Sens) expression is normal in the posterior compartment (delimited by GFP expression), similarly to the anterior compartment that serves as control. Scale bar: 50 μm. **h** Fluorescence intensity profile of NRT-wg and Sens a (grey value; arbitrary units) along yellow lines in the anterior (ant) and posterior (post) compartment of the wing imaginal disc. WT, wild-type. Scale bar: 200 μm. Images and fluorescent intensity profiles are representative of at least 5 images of wing imaginal discs analysed in 3 different experiments. For **a**, **c**, **f**, and **g** white dashed line delimitates posterior and anterior compartment

## Discussion

In this study, we have identified a role for *Sfrp3* in regulating mammary epithelial development during puberty. We found that the absence of *Sfrp3* leads to alterations of MG development, with ducts showing accelerated ductal invasion throughout the fat pad and precocious tertiary branching. Loss of *Sfrp3* leads to luminal cell defects, bringing to alterations of the ductal architecture and integrity. $Sfrp3^{-/-}$ luminal epithelial cells show increased proliferation, with no change in the apoptosis level, loss of polarity features and arrangement into multiple layers protruding and occluding the ductal lumen.

Furthermore, $Sfrp3^{-/-}$ luminal epithelial cells produce milk proteins, consistently with the observed increased tertiary side branching that is typical of pregnancy. Maturation of alveolar precursor cells and differential induction of milk protein genes are mainly controlled by prolactin signalling, which is mediated by JAK2 and STAT5A/B[31,32]. Our bioinformatic analysis of NGS data suggests prolactin signalling as a potential pathway acting in $Sfrp3^{-/-}$ epithelial cells (Fig. 7d). Future work will be needed to better characterise the role of *Sfrp3* in the regulation of prolactin signalling.

Mammary stroma regulates the development and differentiation of epithelial cells, defining as well the side branching pattern[33,34]. This is partially due to a stromal–epithelial communication by secreted factors acting on either an autocrine or paracrine manner. A hallmark of MG development is the bidirectional activity of these factors, which complicates the identification of the source and target cells. Here we observed that *Sfrp3* is expressed in both compartments, but with eight-fold more expression in the stroma. Furthermore, we showed that loss of stromal *Sfrp3* is responsible for the ductal phenotype observed in $Sfrp3^{-/-}$ MG. It is tempting to speculate that the signals from $Sfrp3^{-/-}$ stroma are instructive since they can target transplanted wild-type epithelial cells and participate in their differentiation into multi-layered cells abnormally proliferating and producing milk proteins. Recent studies have unravelled new molecules responsible for the stromal-epithelial communication in mammary epithelial morphogenesis[35,36]. We propose SFRP3 as a mediator of the mammary stromal–epithelial crosstalk.

The Sfrp proteins are traditionally considered the largest family of secreted Wnt inhibitors. Pioneering studies showed that SFRP3 binds Wnt1 and Wnt8, and inhibits canonical Wnt signalling in *Xenopus* embryo and cultured cells[12,37]. However, we demonstrated that SFRP3 acts as a transporter a diffusor of Wg in *Drosophila*, not an inhibitor. We thus hypothesised that *Sfrp3* controls MG development by modulating Wnt signalling at the stromal–epithelial boundary (Fig. 8). Indeed, secreted SFRP3 binds to Wnt2 through its NRT domain, and we detected a substantial increase in Wnt2 accumulation at this boundary and the activation of the Wnt-βcatenin pathway in pubertal $Sfrp3^{-/-}$ whole MGs. Wnt-2, Wnt-5a and Wnt-7b are strongly expressed during ductal development, in the TEB microenvironment, and they seem to be regulators of mammary branching[38,39]. Indeed, the ectopic expression of Wnt2 results in a hypermorphic branching phenotype[40].

Furthermore, we found that *Wnt2* is expressed by stromal fibroblast in the MG, and previous studies showed that *Wnt2* expression accumulates preferentially in the stromal–epithelial boundary[38]. Thus, our current data suggest that SFRP3 might act in the MG as an extracellular transporter of Wnt ligands enhancing their diffusion across cellular compartments. Sfrp3 and Crescent have been shown to promote the diffusion of Wnt8 and Wnt11 in *Xenopus* embryo[41]. In line with this, in the presence of SFRP3, Wg displays a more diffuse distribution at the dorsoventral boundary of the *Drosophila* wing imaginal disc. We propose that SFRP3 binds to stromal Wnt2 in order to diffuse this ligand function to epithelial cells. In the absence of SFRP3, Wnt2 would accumulate in the stroma, resulting in an altered communication with the epithelium, and a defective mammary epithelial morphogenesis (Fig. 8). Future studies will be required to dissect the complex scenario of stromal–epithelial communications better.

Similarly to *Sfrp3*, loss of *Sfrp1* also leads to excessive side branching with clear lobuloalveolar development in the mouse mammary gland[42]. It is to note that, although loss of either *Sfrp1* or *Sfrp3* results in a similar mammary phenotype, distinct molecular routes are activated in their absence at signalling level in order to prompt mammary modifications. For example, in contrast to $Sfrp3^{-/-}$ mice, *Wnt4* and its downstream effector RANKL mRNA levels are increased in $Sfrp1^{-/-}$ mammary gland in comparison to wild-type glands[42]. Additionally, *Sfrp1* expression is found mostly in basal/myoepithelial cells that lack *Sfrp3* expression[43,44]. Lastly, previous work demonstrated that Sfrp1 and Sfrp2 control retinal neurogenesis in a Wnt-independent manner[45]. These results altogether suggest that *Sfrp1* and *Sfrp3*, participate in alternative signalling cascades during MG development. It will be interesting to address the function of other members of the Sfrp family in light of the recent evidence pointing to a stromal expression of *Sfrp2* and *Sfrp4* in both normal and tumour samples[16,46–48].

How is MG development regulated downstream of Wnt signalling? Previous studies reported that mouse models with the activated expression of β-catenin display a precocious

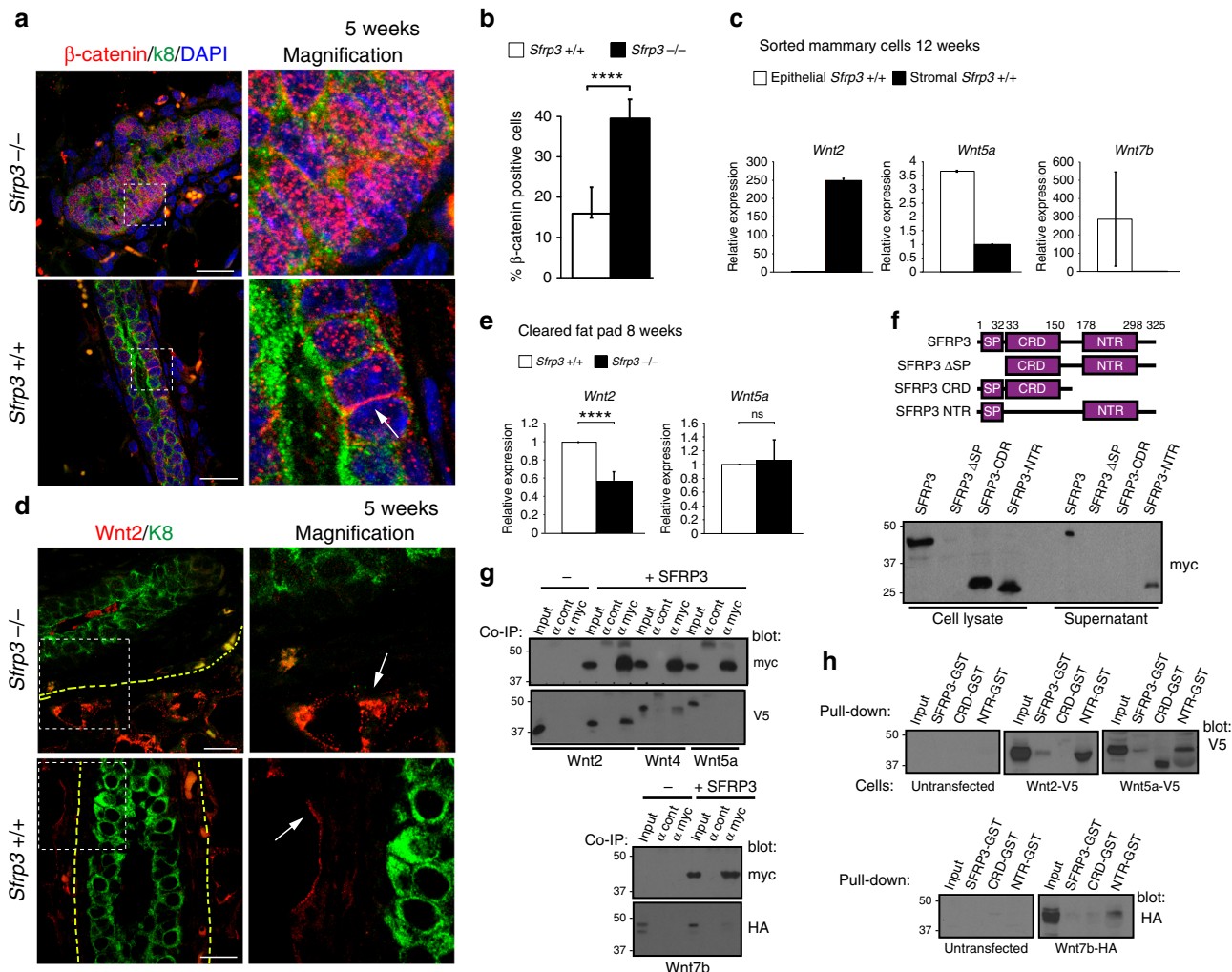

**Fig. 6** *Sfrp3* modulates Wnt signalling in the mammary gland. **a** Representative images of immunofluorescence staining for β-catenin (red), Troma (green) and nuclei (blue), on 5-weeks-old MG. Arrow denote localisation of β-catenin at cellular junctions in controls (Sfrp3$^{+/+}$). Scale bar: 50 μm. **b** Quantification of the number of β-catenin positive cells in mammary ducts of 5 weeks old *Sfrp3*$^{+/+}$ (n = 2, 5 mice) and *Sfrp3*$^{-/-}$ (n = 2, 8 mice) females. Data represent mean ± SD (****p < 0.0001). **c** Relative mRNA level of *Wnt2* (left), *Wnt5a* and *Wnt7b* (middle) and *Wnt16* (right) in whole MG from *Sfrp3*$^{+/+}$ and *Sfrp3*$^{-/-}$ females at 10 weeks of age. n = 6. **d** Representative images of immunofluorescence staining for Wnt2 (red), K8 (green), on 5-weeks-old MG. Arrow denote localisation of Wnt2 at stromal-epithelial boundary. Images are representative of 3 MG sections analysed in 3 different experiments. Scale bar: 20 μm. **e** Relative mRNA level of *Wnt2* (left) and *Wnt5a* (right) in de-epithelialised cleared fat pad from *Sfrp3*$^{+/+}$ and *Sfrp3*$^{-/-}$ females at 8 weeks of age. n = 4. **f** Left, schemes of wild-type myc-tagged SFRP3 and truncated variants. Numbers indicate the aa position of each conserved domain according to UniProtKB database. SP, signal peptide; CRD, cysteine-rich domain; NTR, netrin-like domain. Right, western blot detection of SFRP3-myc in cell lysates and supernatants of MDCKII expressing different variants. The lack of SP domain abolishes protein expression. SFRP3-NRT variant is expressed and secreted, while SFRP3 CDR is expressed but not secreted in the medium. **g** Immunoprecipitation of SFRP3-myc followed by immunoblot with anti-myc, anti-V5 and anti-HA antibody in HEK293 cells transiently co-transfected with wild-type human SFRP3 construct and Wnt2-V5, Wnt5a-V5 and Wnt7b-HA constructs. αcont, cell lysate incubated with negative antibody conjugated beads; αmyc, cell lysate incubated with anti-myc antibody conjugated beads. **h** Pull-down of GST-tagged forms of SFRP3 followed by immunoblot with anti-V5 and anti-HA antibody in HEK293 cells transfected with empty, Wnt2-V5, Wnt5a-V5 and Wnt7b-HA constructs. αcont, cell lysate incubated with negative antibody conjugated beads; αmyc, cell lysate incubated with anti-myc antibody conjugated beads. For **c**, **d**, **e**, data represent mean ± SD (***p < 0.001, ****p < 0.0001, ns, not significant). **f**–**h** blots are representative of 3 different experiments. Source data are provided as a Source Data file

lobuloalveolar development[49], similar to the *Sfrp3*$^{-/-}$ phenotype. We, therefore, expected that in the absence of stromal *Sfrp3* canonical Wnt signalling would be activated. Indeed, we observe nuclear accumulation of β-catenin in *Sfrp3*$^{-/-}$ whole MGs at early puberty, when the levels of *Sfrp3* are higher in the wild types. Besides, the NGS analysis reveals an increase in the expression level of β-catenin target genes *LEF1* and *cMYC* in *Sfrp3*$^{-/-}$ mammary epithelial cells compared to control. These data suggest an early *Sfrp3-Wnt* transcriptional response in a specific epithelial cell population in the MG[50].

The Wnt pathway has emerged as a key signalling cascade participating in mammary organogenesis and oncogenesis[7]. Our NGS data analysis shows that the gene expression profile of *Sfrp3*$^{-/-}$ epithelial cells is associated with BC. Indeed, the hyperproliferation and disruption of epithelial apical polarity phenotype that we observed in *Sfrp3*$^{-/-}$ MG could be reasonably considered as the first steps of a tumorigenic process. Despite this evidence, *Sfrp3*$^{-/-}$ mice did not develop any evident mammary tumour lesion or palpable mass. Also, we did not detect any aberrant collagen deposition and organisation, features that can

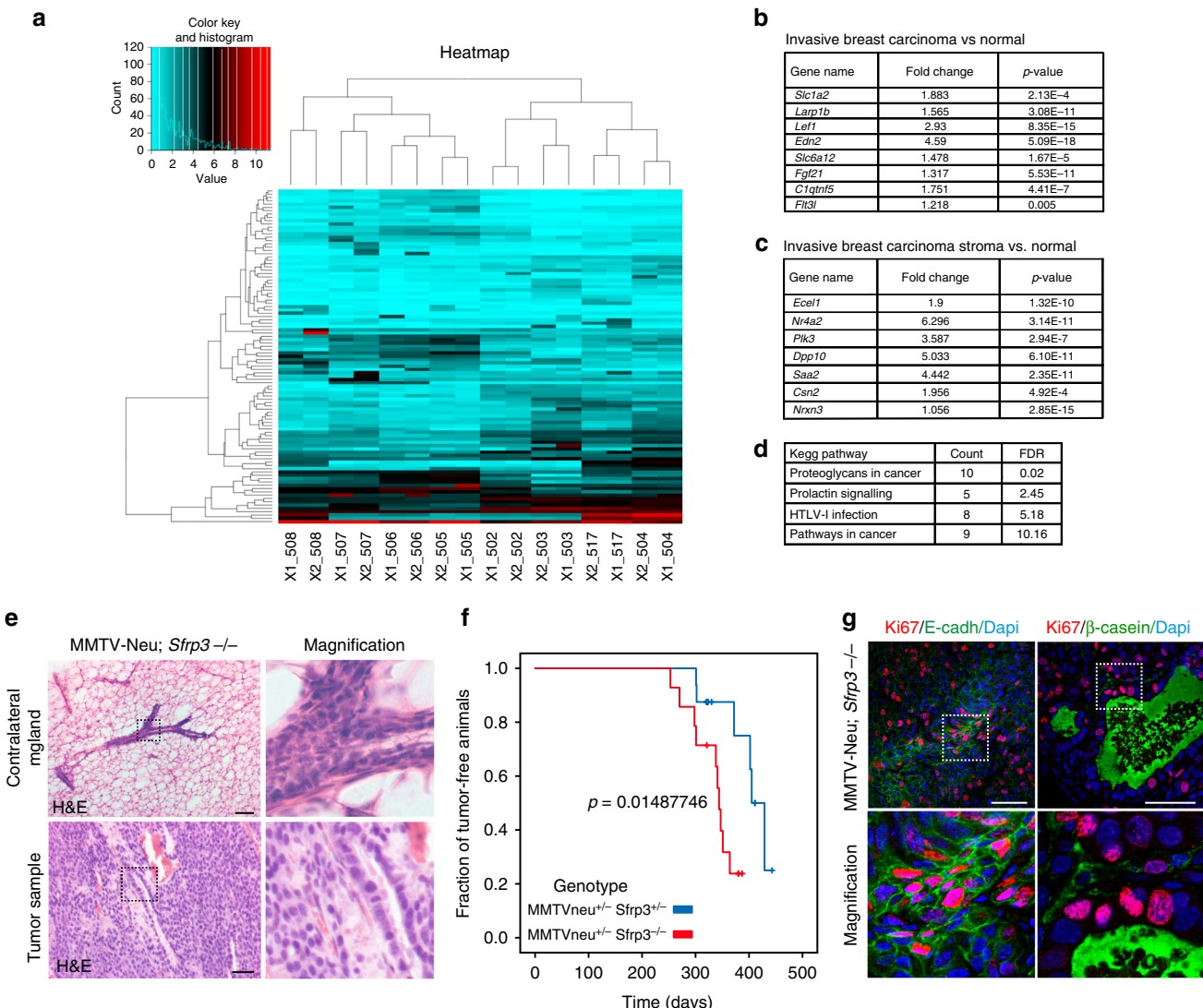

**Fig. 7** *Sfrp3*$^{-/-}$ epithelial cells show a cancer-associated profile. **a** Clustered heatmap of the top 100 DEGs between FACS-sorted *Sfrp3*$^{+/+}$ and *Sfrp3*$^{-/-}$ epithelial cells. Red, increased expression; turquoise, reduced expression. The names of the samples are shown below the heatmap. *Sfrp3*$^{-/-}$ samples: 508, 507, 506, 505. *Sfrp3*$^{+/+}$ samples: 502, 503, 517, 504. Two technical replicates for each sample library were sequenced (X1/X2). **b** Oncomine analysis of the top 100 DEGs. List of genes interrogated in the Cancer Genome Atlas (TCGA) dataset and found to be uregulated ($p \leq 0.005$) in invasive breast carcinoma compared to control sample. **c** Oncomine analysis of the top 100 DEGs. List of genes interrogated in the Finak dataset and found to be uregulated ($p \leq 0.005$) in the stroma of invasive breast carcinoma compared to control sample. **d** KEGG pathway analysis of 102 DEGs. The top 4 pathways with a false discovery rate (FDR) $\leq 10\%$ are shown. **e** Representative images of HE-stained mammary glands from MMTV-Neu; *Sfrp3*$^{-/-}$ virgin females presenting tumours (bottom panels). Upper panels show the contralateral mammary gland used as a control, that present filled ducts. Scale bar: 50 µm. **f** Quantitative analysis of the fraction of tumour-free animals in a MMTV-Neu$^{+/-}$; *Sfrp3*$^{-/-}$ genotype (red line) compared to controls (MMTV-Neu$^{+/-}$; *Sfrp3*$^{+/-}$). $p = 0.01487746$. **g** Representative images of immunofluorescence staining for Ki67 (red), E-cadherin (green), nuclei (blue) (left panels); and Ki67 (red), β-casein (green), nuclei (blue) (right panels), on MMTV-Neu$^{+/-}$; *Sfrp3*$^{-/-}$ virgin females presenting tumours. White square denote area magnified on bottom panels. Scale bar: 50 µm. Images in **e**, **g** are representative of at least 4 MG sections analysed in 4 different experiments. See also Tables S1, S2, S3, S4. Source data are provided as a Source Data file

initiate a process of mammary tumour growth and invasion[51]. We hypothesise that the development of BC requires other genetic and epigenetic changes. Indeed, mutations in canonical Wnt genes, leading to abnormal activation of Wnt signalling, are not sufficient to generate tumours on mouse models, but they require cooperating mutations in order to result in preneoplastic lesions[52]. We identified that lack of SFRP3 in MMTV-Neu animals induces the precocious appearance of premalignant lesions (Fig. 7), suggesting a role for Sfrp3 as a relevant marker for BC susceptibility.

In summary, we have demonstrated that *Sfrp3* is a regulator of MG morphogenesis, being central to the communication between stromal and epithelial cells. Stromal *Sfrp3* constrains epithelial invasion and branching during puberty and controls luminal cells proliferation and differentiation state, likely through the fine-tuning of stromal–epithelial Wnt signalling (Fig. 8). The generation of mouse models with specific loss of *Sfrp3* in different mammary cell compartments will be of extreme value to gain insights on the communication pathways driving MG development.

## Methods

**Mouse lines**. Generation of *Sfrp3*$^{-/-}$ mice in Swiss/CD1 background[14] were a kind gift from Prof F.P. Luyten (KU Leuven, Belgium) and were maintained for our studies on pure CD1 background, as well as *Sfrp3*$^{+/+}$ mice. Virgin female mice

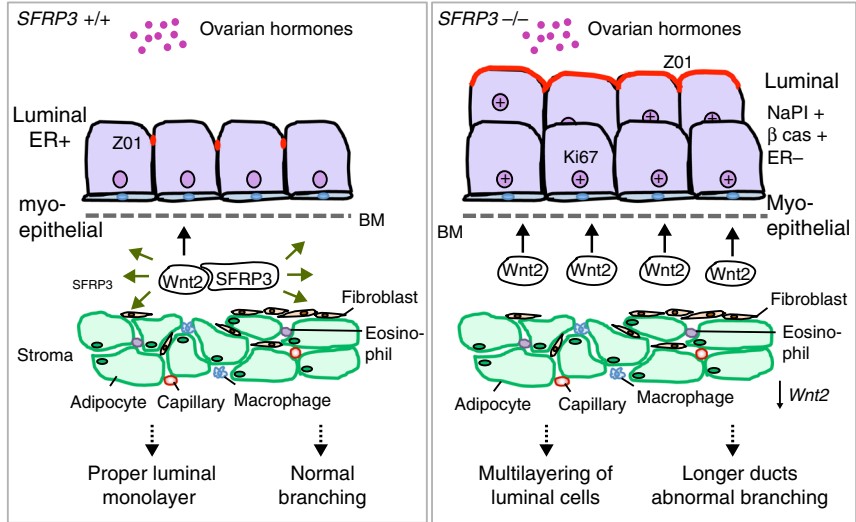

**Fig. 8** Model diagram of how stromal *Sfrp3* regulates MG epithelial morphogenesis. In wild-type mammary gland SFRP3$^{(+/+)}$ secreted by the stroma binds to Wnt2, likely contributing to diffuse the high level of stromal *Wnt2* expression at the boundary. The crosstalk between stroma and epithelium is correct, allowing luminal cells to properly polarise and organise as a monolayer. Ducts branch appropriately during puberty. In the absence of *Sfrp3*$^{(-/-)}$ stromal level of *Wnt2* concentrates at the boundary. The stromal–epithelial communication is altered, resulting in altered arrangement of luminal cells, that are less polarised, overproliferate and produce β casein. Total levels of *Wnt2* expression is reduced. Overall ducts are longer and hyperbranched. BM, basement membrane

were sacrificed at 3, 6, 8 and 10 weeks of age for mammary gland developmental studies. At the end of puberty (8 weeks of age) the oestrus cycle was monitored daily by vaginal smear twice per day (early morning and late afternoon). Only age-matched wild-type and *Sfrp3*$^{-/-}$ females with matching oestrus cycle were used for studies. For pregnancy stages, mice were mated and inspected for the presence of vaginal plug the following morning. The day of the vaginal plug was counted as day 0.5 of pregnancy. The first day after delivery was considered as lactation day 1 (L1). For weight recording, litters were normalised at 7 pups at birth, and weight was measured at day 11 and day 21 (weaning). Male mice were sacrificed at 6 weeks and 6 months of age for mammary gland studies. For transplantation experiments we used CD1 mice and immunocompromised mice. NOD.Cg-*Prkdc*$^{scid}$ *Il2rg*$^{tm1Wjl}$/SzJ stock # 005557 (NSG mice, Jackson Laboratories, Maine, USA) were bred and housed in the CIB animal facility. For analysis of tumorigenesis the *Sfrp3*$^{-/-}$ mice were crossed with FVB/N-Tg(MMTVneu)202Mul/J mice (MMTVneu$^{+/+}$) mice (Jackson Laboratories, Maine USA); the F1 (*Sfrp3*$^{+/-}$ MMTVneu$^{+/-}$) were backcrossed to *Sfrp3*$^{-/-}$ mice. Genotypic analyses allowed to identify the *Sfrp3*$^{-/-}$ MMTVneu$^{+/-}$ and *Sfrp3*$^{+/-}$ MMTVneu$^{+/-}$ genotypes. Appearance of mammary tumours in these animals was analyzed twice weekly, to determine the fraction of tumour-free animals over time.

All mice were housed and bred in a specific pathogen-free (SPF) barrier facility with a 12-h light cycle and chow and water ad libitum. All experiments were approved by the CBMSO's Institutional and Spanish Regional Committees on Ethics of animal research and were in adherence to the NIH "Guide for the Care and Use of Laboratory Animals". Investigators were not blinded to genotype group allocations.

**Fly stocks**. The *Gal4* drivers for ectopic expression experiments using the Gal4/UAS system were *hh-Gal4* (FBti0017278) and *nub-Gal4* (FBti0016825). The expression of *hh-Gal4* is restricted to the posterior compartment of all imaginal discs and the expression of *nub-Gal4* occurs in the entire wing pouch and hinge. The UAS line UAS-GFP (FBti0012493), together with the *Gal4* lines, were obtained from Bloomington Stock Centre (Indiana University, Bloomington, USA).

**Antibodies and reagents**. Monoclonal anti-E-cadherin (#610181) antibody was obtained from BD Transduction Laboratories. Polyclonal anti-Keratin 14 (#PRB-155P) antibody was obtained from BioLegend. Monoclonal anti-Keratin 8 (TROMA-I), anti-ZO1 (R26.4C), anti-Wingless (4D4) and anti-Cut (2B10) antibodies were obtained from Developmental Studies Hybridoma Bank. The rabbit anti-NaPi-11b sera was a kind gift of Dr J. Biber (Universityof Zurich, Zurich, Switzerland) and was used 1:400.

Polyclonal anti-ß casein (M-14, sc-17971) antibody, and anti-ERα (F10. Sc-8002; MC-20, sc-542) was obtained from Santa Cruz Biotechnology. Polyclonal anti-Cleaved Caspase 3 (#9661) antibody was obtained from Cell Signalling. Monoclonal anti-CD326 (EpCAM)-PE (#130-102-967), anti-CD45-FITC (#130-102-778) and anti-CD49f-APC (#130-100-147) antibodies were obtained from Miltenyii Biotec. Monoclonal anti-Ki67 (clone SP6, #MA5-14520) antibody and polyclonal anti-c-myc antibody for staining on *Drosophila* (#PA1-981) were

obtained from Thermo Fisher Scientific. Mouse monoclonal anti-myc tag (clone 4A6, cat#05-724) antibody for IP and western blot, and anti-active b-catenin (8E7) was obtained from Merck-Millipore. Polyclonal anti-V5 tag (#GTX117997) was purchased from GeneTex. Polyclonal anti-HA tag (#H6908) antibody was obtained from Sigma. Alexa Fluor 488/555/647-conjugated antibodies were obtained from Thermo Fisher. Matrigel Growth Factor reduced (#356230) was obtained from BD Biosciences. Chambered coverglasses (#155383) were obtained from LabTek. RNeasy Plus Mini kit (#74134) was from Qiagen. ThermoScript RT-PCR System (#11146-024) was obtained from Invitrogen. High Capacity RNA-to-cDNA kit (#4387406) was from Thermo Scientific. GoTaq qPCR Master Mix (#A6002) was obtained from Promega.

**Genotyping PCR**. Mice were genotyped by PCR analysis on genomic DNA extracted from tail tip. Two combinations of primers were used, one specific for the *Sfrp3*$^{+/+}$ allele (primers *Sfrp3*$^{+/+}$ allele forward and reverse) and the other specific for the *Sfrp3*$^{-/-}$ allele (primers *Sfrp3*$^{-/-}$ allele forward and reverse). MMTVneu$^{+/+}$ transgenic mice were genotyped by PCR using the primers oIMR0386 TTTCCTGCAGCAGCC TACGC and oIMR0387 CGGAACCCACATCAGGCC to amplify the transgene, using as internal control the amplification with the primers oIMR8744 CAAATGTTGCTTG TCTGGTG and oIMR8745 GTCAGTCGAGTGCACAGTTT, following the conditions described by the provider (The Jackson Laboratory).

**Whole mount analysis of mammary glands**. Abdominal (no. #4) glands were dissected from mice and spread on glass slides. After fixation in Carnoy's fixative (60% ethanol, 30% chloroform, and 10% glacial acetic acid) for either 2–4 h at room temperature or overnight at 4 °C, they were hydrated and stained with Carmine alum overnight at room temperature. Stained mammary glands were dehydrated, cleared in xylene overnight and mounted with Permount mounting medium. Images were taken with a Nikon D750 digital camera and processed with ImageJ.

For each gland, the ductal length was measured as the distance between the lymph node and the distal end of the longest duct. Branching was quantified by manually counting the total number of branch nodes within three fields either adjacent to the sides of the lymph node for mammary gland or randomly distributed for transplanted mammary gland.

**Histological staining and analysis**. For morphological analysis, carmine-stained mammary glands were cleared in xylene, embedded in paraffin, sectioned at 5 μm and stained with hematoxylin and eosin according to standard procedures. Images were collected on a Zeiss Axioskop2 plus microscope equipped with a colour charge-coupled device camera. Ducts with filled lumen were manually counted in 5–7 images per mammary section and expressed as a percentage of the total number of ducts per image.

For collagen staining, mammary glands were fixed with 4% PFA overnight, embedded in paraffin, sectioned at 5 μm and stained with PicroSirius red. Images of Sirius red-stained sections were taken with a Nikon Eclipse TE2000-U microscope under bright and polarised light and processed by using Image J software. The

number of the different coloured collagen fibres was measured as previously described[53]. Briefly, we subtracted the black-component image from each polarised image in order to eliminate interstitial space and non-collagen elements that are birefringent materials as collagen. Within the subtracted image we determined the hue (colour) of each pixel by using the following hue definitions: red 1–9 and 130–255; orange 10–38; yellow 39–51; green 52–128; interstitial space 129–229. The number of pixels within each hue range per image was calculated and expressed as a percentage of the total number of collagen pixels. The proportion of thick collagen fibres per image is given as the sum of red, orange and yellow pixels while the proportion of thin collagen fibres is due to green pixels percentage. Eleven fields per mammary gland were imaged.

For the quantification of collagen content the total number of collagen pixels (corresponding to hue range 1–255) was expressed as a percentage of the total number of pixels per image. Eleven images per mammary gland were analysed. The level of anisotropy of collagen fibres was measured by using the FibrilTool Image-J plug as previously described[54]. For each image a ROI (region of interest) was selected including one single mammary duct and the surrounding area of collagen. A score for anisotropy was assessed for each ROI: 0 for no order (purely isotropic arrays) and 1 for perfectly ordered, i.e., parallel fibrils (purely anisotropic arrays). Between six and eleven images per mammary gland were analysed.

**Immunofluorescence on mammary gland sections.** Carmine-stained mammary glands, after whole-mount pictures had been taken, were cleared in xylene, embedded in paraffin and sectioned at 5 μm. Sections were deparaffinised and rehydrated through a graded series of ethanols. Antigen retrieval was performed by boiling sections in 10 mM sodium citrate buffer (pH 6) for 15 min. The sections were blocked in PBS + 3% BSA and then incubated with the primary antibodies anti-K14 (1:200), anti-K8 (1:2), anti-E-cadherin (1:400), anti-ZO1 (1:500), anti-CC3 (1:500), anti-Ki67 (1:500), anti-ERα (1/200), anti-b-catenin (1/100), anti-NaPI (1/250) diluted in PBS + 3% BSA overnight at 4 °C. After 3 washes in PBS, the secondary antibodies rabbit 555 (1:500), mouse 488 (1:500), rat 555/647 (1:500), plus nuclei marker DAPI (1:500) were applied in PBS + 3% BSA for 1 h at room temperature. When anti-β casein (1:1000) and goat 555 (1:500) were used as primary and secondary antibody, respectively, blocking was done with PBS + 5% donkey serum while the primary and secondary antibody solutions were PBS + 1% donkey serum. Sections were mounted with Mowiol mounting medium and imaged using a Zeiss Confocal LSM710 microscope. Ki67 positive cells were manually counted in 4–5 images per mammary section and expressed as a percentage of the total number of cells per image.

**Clearing of mammary fat pad.** For RNA extraction from three-weeks-old cleared fat pad, mice of that age were sacrificed and the entire right abdominal mammary gland was excised. The region between the nipple and the lymph node, containing the endogenous epithelium, was separated from the fat pad (cleared) by performing a cut just after the lymph node. The tissue between the nipple and the lymph node was spread on a glass slide, fixed and stained with Carmine alum in order to confirm that the entire rudimentary tree was removed. The cleared fat pad was collected in RNA later (Invitrogen) and either stored at −80 °C or immediately processed for RNA extraction.

For RNA extraction from 8-weeks-old cleared fat pad, 3-weeks-old mice were anaesthetised and the right abdominal mammary gland was exposed through a small mid-sagittal cut in the skin of the lower abdomen and oblique cuts along the right hind leg. The connection between the abdominal (no. #4) and the inguinal (no. #5) gland was cauterised to prevent ingrowth from the 5th gland. The endogenous epithelium was eliminated by removing the tissue between the nipple and the lymph node. The removed tissue was stained with Carmine alum to check for cleared margins. After clearing of the mammary fat pad, the incision sites were closed and mice were sacrificed 5 weeks after surgery. The right cleared mammary fat pad was processed for RNA extraction while the left abdominal gland (not operated) was analysed by whole mount carmine alum staining to check for proper epithelial ducts development.

**Mammary epithelial transplantation.** Transplantation of epithelial ducts was performed according to the standard procedure previously developed[55]. Three weeks old females were used as recipients and were anaesthetised. The region between the nipple and the lymph was excised from the abdominal mammary gland. Eight weeks old females were used as donors of epithelium. Small pieces of the abdominal mammary glands (1 mm, close to the lymph node) were cut and placed into DMEM/F12 media buffered with HEPES and NaHCO3, 10% foetal bovine serum (FBS) for the time of the surgery. One piece of donor epithelium was freshly transplanted into the centre of the cleared fat pad of the recipient female. When CD1 mice were used as recipients, the transplantation was performed only in one mammary gland (right) while the contralateral (left) was left intact and used as control. When NSG mice were used as recipients, the fat pad of both abdominal glands was cleared. Sfrp3[+/+] and Sfrp3[−/−] donor epithelium were transplanted into the right and left cleared fat pad, respectively, of the same recipient female. The transplants were analysed 5 weeks afterwards.

**Ovariectomy.** Both ovaries were removed from anaesthetised 3-weeks-old females by performing two separate lateral incisions in the lower back. Five weeks after surgery, females were sacrificed and mammary glands were harvested for whole mount carmine alum staining.

**3D culture of mammary organoids and staining.** We purified mammary organoids from CD1 mice following the protocol previously optimised and described for FVB mice[56]. Mammary glands pairs #3, #4 (without lymph node), and #5 were harvested from virgin 8 weeks old females. They were minced and digested in collagenase buffer (DMEM/F12 medium with 2 mg/ml collagenase, 2 mg/ml trypsin, 5% FBS, 5 μg/ml insulin, and 50 μg/ml of gentamicin) for 30–40 min at 37 °C. Samples were centrifuged at 1500 rpm for 10 min. To recover additional epithelial tissue, the top fatty layer was resuspended in DMEM/F12 and centrifuged again. The epithelial pellets were resuspended in DMEM/F12 and treated with 40 μl DNase I (2 U/μl) for 3 min. They were then resuspended in DMEM/F12 and centrifuged at 1500 rpm for 10 min in order to obtain a compact red pellet. After four rounds of differential centrifugation (quick spin at 1500 rpm) the final white pellet consisted mostly of organoids. Organoids were embedded in Matrigel Growth Factor Reduced at 2 organoids/μl and plated as 75/μl suspensions in 4-well Chambered coverglass over a 37 °C heating block. Gels were allowed to polymerize for 50 min at 37 °C, 5% CO2, and then overlaid with organoid medium (DMEM/F12 with 1% penicillin-streptomycin and 1% ITS). The medium was changed after 2 days and organoids were analysed after 6 days of culture. They were fixed with 2% PFA (15 min light shaking) and permeabilized with DPBS + 0.5% TritonX-100 for 1 h at room temperature. After blocking with DPBS + 10% FBS + 1% BSA for 2 h at room temperature, organoids were incubated overnight at 4 °C with antibody solution (DPBS + 1% FBS + 1% BSA) plus the primary antibody anti-K14 (1:2000). Organoids were washed with DPBS + 10% FBS (3×, 10 min each) and incubated for 1 h with antibody solution plus the secondary antibody rabbit 488 (1:500), phalloidin-555 (1:500), and DAPI (1:500). After three washes with DPBS, the chamber was removed by cutting it with a razor blade and the glass bottom coverslip was overlaid with PBS for imaging. A W-Plan Apochromat 40×/1.0 W objective lens (Carl Zeiss) in a laser scanning confocal microscope (Zeiss LSM 710) was used for high magnification image acquisition.

Organoid size was quantified by measuring the longest axis of the organoid while lumen size was quantified by measuring the longest axis of the central cavity. Six to eighteen organoids were analysed per female.

**Isolation of primary mammary cells by FACS.** Mammary glands pairs #2, #3, #4, and #5 were isolated from virgin 12-weeks-old females. They were chopped and incubated in the dissociation solution (DMEM/F12 medium with 10× gentle collagenase/hyaluronidase, 5% FBS and 50 μg/ml of gentamicin) overnight at 37 °C, 5% CO2. After dissociation, cells were washed in HBSS + 2%FBS and centrifuged at 400 × g for 5 min. The pellet was incubated in HBSS + 2%FBS + NH4Cl (mixture 1:4) for 10 minutes for lysis of red blood cells and centrifuged at 400×g for 5 min. The resultant pellet contains epithelial cell organoids as well as stromal cells and lymphocytes. To obtain a single-cell suspension, the pellet was gently pipetted for 3 min in 2 ml of trypsin-EDTA 0.25%, centrifuged, and then pipetted for 3 min in 2 ml Dispase II (5 mg/ml) and 200 μl DNaseI (1 mg/ml). After resuspension in HBSS + 2% FBS, cells were passed through a 40 μm cell strainer filter in order to dissociate cell clumps and centrifuged. The pellet was blocked in FACS buffer (PBS pH7 + 1 mM EDTA + 25 mM HEPES) + 10% FBS for 10 minutes and then incubated for 10 minutes with the fluorescently labelled primary antibodies anti-CD45, anti-CD326 (EpCAM), and anti-CD49f. Cells were washed from unbound antibodies and immediately sorted on a BD FACS Aria Fusion Becton Dickinson California flow cytometer (nozzle size 85 μm, sheet pressure 45 psi) equipped with DIVA software. Cell viability was determined by using DAPI. Cell clusters and doublets were electronically gated out.

**Constructs generation.** Human wild-type SFRP3 was expressed from pcDNA 3.1 myc-His (−) cloning vector (Invitrogen, Carlsbad, CA) (primers hSfrp3 used for cloning). Truncated variants of myc-tagged Sfrp3 were obtained by PCR on human wild-type Sfrp3 insert. The following primers were used: Sfrp3 ΔSP for generating SFRP3 variant lacking the signal peptide sequence (SFRP3 ΔSP); Sfrp3 ΔCRD (#1 + #2) for generating SFRP3 variant lacking the CRD domain (SFRP3 ΔCRD); Sfrp3 ΔNTR for generating SFRP3 variant lacking the NTR domain (SFRP3 ΔNTR). Active Wnt2-V5 (Addgene plasmid #43809), Wnt4-V5 (Addgene plasmid #43812) and Wnt5a-V5 (Addgene plasmid #43813) plasmids were a gift from Xi He. Wnt7b-HA construct was kindly provided by Paola Bovolenta's lab. For injection in Drosophila, mouse Sfrp3 cDNA was fused in frame to a C-terminal myc tag, cloned into a pUAST vector[57] and then inserted in the attP site 3R (86Fb) to generate a transgenic fly line expressing SFRP3 under the UAS promoter. UAS-Sfrp3 was expressed using hh-Gal4 or nub-Gal4 drivers.

**SFRP3 expressing stable clones and western blot.** MDCKII cells were grown in MEM + 5% FBS. MDCK cells stably expressing either wild-type or truncated SFRP3 variants were selected for 10 days using G418 (0.5 mg/ml). Stable clones were grown in M12 wells plate until reaching confluency and then cultured in serum-free MEM for 12 h. After that, supernatants were collected and cells were

lysed in 200 μl of Laemmli 2× sample buffer. Supernatants were centrifuged at 2000 rpm for 5 min to remove cell debris. The pellet was discarded and 4 volumes of cold 100% acetone were added to the supernatant. Samples were incubated at −80 °C for 10 min and centrifuged at 15,000 rpm for 15 min at 4 °C. The supernatant was discarded and the pellet was dried for 15 min at room temperature. Proteins were then resuspended in 400μl of Laemmli 2× sample buffer (Sigma Aldrich). A total of 15 μl of cell lysate and 40 μl of medium were loaded on 10% SDS-PAGE. Transblotted nitrocellulose membranes were incubated with anti-myc antibody (1:500).

**Immunoprecipitation**. HEK293 cells were grown in MEM + 5%FBS. The day before transfection 2–2.2 × 10^6 cells were seeded in a P100 plate. Medium was changed 1–3 h before transfection by Calcium Phosphate performed as following. 6.25 μg of the construct expressing the wild-type variant of human myc-tagged SFRP3 and 18.75 μg of the construct expressing active V5/HA-tagged Wnt ligand were mixed in water plus 61 μl of 2 M CaCl₂. The DNA mix was added to 500 μl of 2× HBS pH7, incubated for 30–60 s, and added dropwise to cells. The following day cells were lysed in 1 ml lysis buffer (150 mM NaCl, 50 mM Tris pH 8, 1% NP40, protease inhibitors). After clearing with unconjugated Sepharose beads, half of the lysate was incubated with anti-myc conjugated beads, and the remaining half was incubated with a negative control antibody conjugated beads for 2 h at 4 °C. After three washes with lysis buffer and one wash with PBS, the pellet was resuspended in 500 μl Laemmli 2× sample buffer, and boiled. Fifteen microliter were loaded on 10% SDS-PAGE and analysed by western blot using anti-myc antibody (1:1000) for detection of SFRP3, anti-V5 antibody (1:1000) for detection of Wnt2 and Wnt5a, and anti-HA antibody (1:500) for detection of Wnt7b.

**RNA extraction and quantitative PCR analysis**. Mouse tissues were mechanically homogenised and total RNA was extracted by Trizol Reagent (Invitrogen) according to the manufacturer's protocols. FACS-sorted cells were immediately lysed in Trizol Reagent and total RNA was purified by RNeasy Plus spin column kit.

For RT-PCR on mouse tissues one μg RNA was reverse transcribed with the ThermoScript RT-PCR system using random hexamers. *Sfrp3* (primers pair #1) and *GAPDH* (as normaliser gene, primers pair #1) were amplified from 2 μl of CDNA. For quantitative Real Time PCR one μg RNA for tissue sample or 600 ng RNA for FACS-sorted cells sample was retrotranscribed to cDNA with the High Capacity RNA-to-cDNA kit. Real Time PCR was performed on 10 ng of CDNA using the GoTaq qPCR Master Mix and 2.5 μM gene-specific primers in a final volume of 10μl using the ABI 7900HT. The genes amplified from mouse tissue were: *Sfrp3* (primers pair #2), *Wnt2*, *Wnt5a*, *Wnt7b*, *Wnt16*, *ß casein*, *Lalba*, *WAP*, *axin2*, *cyclinD1*. The genes amplified from sorted cells were *Sfrp3* (primers pair #2), *Wnt2*, *Wnt5a*, *Wnt7b*, *CD24a*, *CD49f*, *Aldha1a*. *ßactin*, *GAPDH* (primers pair #2) and *Ppia* were used as internal controls. All primer sets were tested for primer efficiency using a six-fold dilution series containing six dilutions. Conditions of Real Time quantitative PCR were as follows: 95 °C for 10 min, then 40 cycles of 95 °C for 15 s and 60 °C for 1 min followed by a dissociation stage. The data were analysed using comparative $C_T$. All qPCR results were obtained from three biological repeats and at least two technical repeats.

**Low-input RNA sequencing**. RNA sequencing libraries were prepared following the SMARTseq2 protocol[58] with some modifications. Briefly, RNA was quantified using the Qubit RNA HS Assay Kit (Thermo Fisher Scientific) and 10 ng of RNA was used for cDNA synthesis. Reverse transcription was performed using SuperScript II (Invitrogen) in the presence of oligo-dT30VN (1 μM), template-switching oligonucleotides (1 μM) and betaine (1 M). The cDNA was amplified using the KAPA Hifi Hotstart ReadyMix (Kappa Biosystems), 100 nM ISPCR primer and 8 cycles of amplification. Following purification with Agencourt Ampure XP beads (1:1 ratio; Beckmann Coulter), product size distribution and quantity were assessed on a Bioanalyzer using a High Sensitivity DNA Kit (Agilent). Two hundred nanogram of the amplified cDNA was fragmented for 10 min at 55 °C using Nextera® XT (Illumina) and amplified for 12 cycles with indexed Nextera® PCR primers. Products were purified twice with Agencourt Ampure XP beads (0.8:1 ratio) and quantified again on a Bioanalyzer using a High Sensitivity DNA Kit.

The Nextera® libraries were sequenced on HiSeq2000 (Illumina, Inc) in paired-end mode with a read length of 2 × 76 bp using TruSeq SBS Kit v4 and TruSeq Dual Index of 8 bp + 8 bp. We generated a mean of 42 million paired-end reads for each sample in a fraction of a sequencing v4 flow cell lane, following the manufacturer's protocol. Image analysis, base calling and quality scoring of the run were processed using the manufacturer's software Real Time Analysis (RTA 1.18.66.3) and followed by generation of FASTQ sequence files by CASAVA.

**Bioinformatics analysis**. More than 3 Gb of sequencing reads were obtained from each sample in fastq format. Quality analyses were performed over reads using FastQC software. Reads were to *M. musculus* reference genome (GRCm38) using tophat2 v2.0.14 with default parameters.

Differential expression analysis was performed using Cuffdiff v2.2.2 package, obtaining the expression values (RPKM) for each gene. A total of 1408 DEGs (RPKM > 1, *q* value < 0.05) were identified. Two datasets (TCGA and Finak) publicly available in Oncomine cancer microarray database were used to examine the expression of the top 100 DEGs in BC tissues. We selected *p* ≤ 0.005 as a threshold. Also, a Functional Enrichment Analysis (FEA) on a subgroup of 102 DEGs genes of interest was performed with an in-house R script that uses the FGNet package.

**Immunostaining on *Drosophila* wing imaginal discs**. Imaginal discs were dissected in PBS, fixed in 4% PFA for 20 min at room temperature. They were blocked in PBS + 1%BSA + 0.3% TritonX-100 for 1 h at room temperature and then incubated with primary antibodies in PBST-BSA overnight at 4 °C, washed four times in blocking buffer and incubated with the appropriate fluorescent secondary antibody for 1 h at room temperature in the dark. They were then washed and mounted in Vectashield mounting medium. Primary antibodies used were: rabbit anti-c-myc (1:500), mouse anti-Wingless 4D4 (1:50), mouse anti-Cut 2B10 (1:100) and guinea pig anti-Sens (1:1000, a gift from Isabel Guerrero), and mouse anti Distalless (dll) (1:1000, a gift from Carlos Estella[59]. Confocal images were captured using an A1R + Nikon confocal microscope.

**Quantification and statistical analysis**. Results are expressed as the mean ± standard deviation (SD). To calculate statistical significance, the two tailed Student's *t*-test was used. Sample size *n* is reported for each experiment in figure legends. *p* value < 0.05 was considered significant. All calculations were performed using Prism software.

## Data availability
The authors declare that all data supporting the findings of this study are available within the article and its supplementary information files or from the corresponding author upon reasonable request. The uncropped and unprocessed scans of the most important blots, and the excel tables of all quantifications can be found in the Source Data file in the Supplementary Information.

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

## Acknowledgements

We thank Carmen M. Ruiz-Jarabo for her comments on the manuscript and members of the Martin-Belmonte lab for helpful discussions. We thank the histology service at the National Centre of Biotechnology (CNB), Madrid, Spain, for technical support. We thank the staff of Confocal and Optical Microscopy Service, Flow Cytometry Unit and Drosophila Transgenesis Service at the CBMSO, Madrid, Spain, for providing excellent assistance and technical support. We acknowledge the Centro de Análisis Genómico (CNAG-CRG), Barcelona, Spain, for the low-input RNA sequencing. We thank the Genomics and NGS Core Facility at the CBMSO (CSIC-UAM, which is part of the CEI UAM + CSIC) Madrid-Spain, for the NGS data analysis (http://www.cbm.uam.es/genomica) and the CIB-CSIC animal facility for the breedings and tumour analyses. We thank the labs of Isabel Guerrero, Jose Felix de Celis and Carlos Estella (CBMSO) for advices, reagents and support for the *Drosophila* experiments. This work was supported by grants from the MINECO (BFU2015-71244-ERC; BFU2014-52125-REDT; BFU2014-57831), and Fundacion Ramón Areces to F.M-B. M.D.B. was supported by a MINECO-FPI 2015 PhD fellowship. M.H. was supported by a MINECO-FPI 2012 PhD fellowship. M.B.F. was supported by a La-Caixa PhD programme fellowship.

## Author contributions

F.M.-B. and I.B. perform all the conceptual and experimental design of this study. I.B., T.G.M. and C.C.G. performed the experiments on mice. M.D.B and R.M. performed the study on *Drosophila*. I.B., M.H., M.B.F. and J.T. performed the in vitro experiments. J.A. G.S. provided NGS mice. J.A.G.S. and S.S. G.-M. supervised mouse transplants and tumorigenicity analyses of the mice. I.B. analysed the experimental data. I.B. and F.M.-B. wrote the manuscript. All the authors read and edited the manuscript.

## Additional information

**Competing interests:** The authors declare no competing interests.

