## [Peer Review File · Nature Communications]

Reviewers' Comments:

Reviewer #1:

Remarks to the Author:

The data demonstrating the role of SFRP3 in mammary development and the fact that it is primarily derived from the stroma is interesting, nevertheless the current manuscript leaves too many questions unanswered, and lacks solid mechanistic data.

The authors demonstrate that SFRP3 is required for proper mammary development, although there is no indication of the expression of the other SFRP genes. Is mammary development solely reliant on SFRP3? What are the relative mRNA levels of the other SFRP genes in the epithelial vs stromal cells? Furthermore what is the expression of the other SFRP members during mammary gland development at 3,6 and 8 weeks? The observation that SFRP3^{-/-} mice results in morphological defect in the ducts, mirrors the results of a study by Gauger et al, 2012, Loss of sfrp1 promotes ductal branching in the murine mammary gland. BMC Dev Biol.

The authors use an in vivo drosophila model to demonstrate that SFRP3 acts as a Wnt antagonist in vivo, while the data presented is quite convincing, the question remains whether in their mouse model whether SFRP3 acts as a canonical or non-canonical antagonist? The authors saw no significant changes in most classical in canonical signaling target genes, and state that it may work in a nonconventional manner. The authors saw changes in Wnt5a expression in SFRP3^{-/-} mice.

Wnt5a is a known modulator of morphogenesis through ROR1, did the authors explore whether the loss of SFRP3 could lead to any changes in Wnt5a binding to ROR1 independent of an SFRP3 interaction? Or even if the loss of SFRP3 could cause an increase in the phosphorylation of Dvl? Does Wnt2 bind to a Frizzled receptor on the epithelial cells to activate Wnt signaling?

In figure 5 the authors show that differential expression of Wnt2, Wnt5a and Wnt7b in the stromal sorted cells vs the epithelial cells in SFRP3^{+/+} mice. Does the expression of these change in SFRP3^{-/-} mice?

The immunoprecipitation data is interesting, demonstrating that the NTR is required for secretion. The authors state SFRP3 "very likely binds through its CRD". This should be confirmed experimentally through co-immunoprecipitation.

The data demonstrates that loss of SFRP3 results in a transcriptomic profile which exhibits a more cancer-associated profile, although they were not able to generate any mammospheres in SFRP null mice and did not report any tumour development in the mouse model. Do tumours develop in these mice in the longer term?

Specific Comments:

1. Title doesn't reveal the subject content. It gives an impression that the study is going to be on normal mammary gland development. Abstract describes both the cancer and mammary gland development.
2. Excellent animal studies. However, clear hypothesis, aims and the connectivity with relevant experimental model is missing.
3. It is known sFRPs are downregulated/dysregulated in cancers. The experimental approaches should have been focused towards mammary gland development/milk secretion/ production rather than reporting for tumour susceptibility.
4. Sfrp3^{+/+} status will be interesting in section "loss of Sfrp3 leads to structural defects of epithelial ducts and precocious differentiation of mammary gland"
5. The method authors used did not completely block estrogen production. Other sources of hormone production should be kept in mind. Did not see any change in estrogen and progesterone in the serum indicating the presence of the hormones from other sources. Cannot comment on effect of hormones on sfrp3 unless it is totally abolished.
6. Although drosophila is a good model to study morphogens, it may not be the biologically closer model to mice, better approaches using cell line or breast cancer cells would have been more

appropriate in section "Sfrp3 controls mammary gland development through the modulation of Wnt signalling"

7. What will be the status of WNTs in sFRP3^{+/+} or the wild type in section "Sfrp3 controls mammary gland development through the modulation of Wnt signalling"

8. The authors showed the interaction in hek293 cells and relating it to mammary stroma. The best approach would be to do IPs-assay on the tissues of mammary gland. They saw a decrease in WNT2 mRNA in sFRP3^{-/-} knock out mice in section "Sfrp3 controls mammary gland development through the modulation of Wnt signalling"

9. Though the authors did not observe tumorigenic properties they cannot conclude sfrp3 does not act as tumor suppressor gene. This should be studied in a mice tumor model targeting cells expressing sfrp^{+/+} in section "Loss of Sfrp3 leads to alteration in the expression level of genes associated with cancer"

10. The authors jumped into human breast cancer gene data set to correlate the mice sfrp3^{-/-} NGS status. Non-tumorigenic expression cannot be compared with tumorigenic expression as the receptors and ligands interaction may vary in section "Loss of Sfrp3 leads to alteration in the expression level of genes associated with cancer"

11. In section "Loss of Sfrp3 leads to alteration in the expression level of genes associated with cancer" above should be studied in a tumor model concluding for anticancer properties.

12. In discussion, the section suggest that stromal SFRP3 binds to stromal Wnt2 to maintain Wnt2 normal expression level needs to be proven.

General Comment: Experimental models should be specific to the questions

The aim should be to address the mammary gland development. Instead, authors are trying to relate mammary gland development during pregnancy to cancer, addressed sfrp 3 interaction with wnts in cell lines but not clearly addressed sFRP^{-/-} knock out mice scenario in puberty or pregnancy or in absence of hormones. Showing evidence for targeting sFRP3 stroma cells would have been a better approach

Reviewer #2:

Remarks to the Author:

Bernascone and colleagues report the study of Sfrp3, a member of a secreted glycoprotein family, as a mediator of cross talk between stroma and epithelium during mammary gland development. Per request from the editors, my review will focus primarily on the part of the study conducted in Drosophila but will also touch upon what I perceive as major weaknesses in the whole report. In Drosophila, the authors overexpressed myc-tagged Sfrp3 in half of the larval wing discs. The GAL4 driver used would result in expression in the columnar epithelial cells. This 'interferes with the symmetrical gradient of Wingless' (Drosophila Wnt1) and reduced the expression of Wg target *Sen*. The authors show that Notch signaling is not affected in the same system, indicating some specificity. While the effect on *Sen* supports the conclusion that Sfrp3 overexpression inhibits Wg, I cannot see the effect on Wg gradient that the authors refer to. The staining still looks quite symmetrical across the D/V boundary to me, and if anything, I see reduced Wg in the P compartment where Sfrp3 is expressed. To be more convincing, the authors need to quantify Wg levels by quantitative image analysis such as a line scan across the D/V boundary. The reduction of Wg by Sfrp3, if real, is puzzling because the literature cited by the authors (ref 11-12) suggest that Sfrp3 homologs inhibit Wnt signaling (by direct binding to Wnt proteins), not Wnt production.

So, it is odd that Sfrp3 in wing discs would reduce the level of Wg. The effect could be indirect, as in blocking the proliferation and reducing the number of Wg-producing cells, for example, and may or may not involve inhibition of Wg signaling. What is even more puzzling is the authors' finding that Sfrp3^{-/-} tissues show decreased expression of some Wnt proteins (Fig.5), opposite of what you expect from the Drosophila data where overexpression of Sfrp3 decreased Wg. Again, the authors' finding is contrary to the literature that Sfrp proteins, as secreted extracellular proteins, would affect Wnt function outside of the cell but not necessarily Wnt expression.

Related, the authors conclude that secreted Sfrp3 binds to Wnt2 (outside the cell) to maintain Wnt2 expression in the same cells. Is there precedent for Wnt signaling acting to maintain Wnt expression? If Wnt2 signaling is really affected, why are targets of Wnt signaling not altered and there was no change in beta-catenin nuclear localization in Sfrp3^{-/-} cells (pg. 13)? The authors discuss Sfrp proteins as inhibitors of Wnt signaling. But if Sfrp3^{-/-} mutation reduces Wnt2 levels, wouldn't the wild type then be an activator of Wnt2?

The major weakness in this report is the use of 'stromal' when it is really 'systemic'. For example, in transplantation experiments in Fig. 4, mutant epithelium developed normally when placed into wild type stroma but wild type epithelium developed abnormally when placed into mutant stroma. The authors conclude that Sfrp3 in the stroma directs normal development, but it is not just the stroma but the whole rest of the mouse that bears the genotype of the stroma in each case. So it could be contribution from another tissue and not the stroma that is important. In another example, Sfrp3^{-/-} stroma is shown to express reduced Wnt2 compared to wild type stroma (Fig. 5F). The authors conclude that Sfrp3 function is 'cell compartment autonomous' in directing Wnt 2 expression. Again, it is not just the stroma that is mutant but the whole mouse is. It could be Sfrp3 function in another tissue that is responsible for Wnt expression in the stroma. To conclude that stromal Sfrp3 is important for epithelial development, tissue-specific knock down/out of Sfrp3 in the stroma is needed.

In sum, the developmental phenotype of Sfrp3^{-/-} mice are extremely interesting. But the data from follow up analyses to get at the mechanism fall short of supporting the conclusions.

Minor point on data presentation; it is potentially confusing to show y-axis values that are so different in Fig. 5 D, E and F, for the same gene. I understand that large values result from very small levels in one sample used to normalize (like dividing by zero), but it would be more informative to normalize relative to a single sample for each gene (e.g. use whole gland Wnt 2 in wild type to normalize all other Wnt2 signals). That would allow comparison across time points and also across tissues.

Reviewer #3:

Remarks to the Author:

This is a very interesting study by Bernasconi et al. that describes a novel role for the secreted Wnt inhibitor, Sfrp3, in regulating branching morphogenesis and differentiation of the mammary epithelium through the stromal-epithelial crosstalk. The authors provide convincing expression data that Sfrp3 predominantly resides in the stroma of the mammary gland and is downregulated upon pregnancy. Genetic loss-of-function of Sfrp3 prompted accelerated ductal extension and branching during puberty and precocious alveolar development in the adult. These phenotypes were accompanied by alterations in luminal polarity, multicellular accumulation, and defects in differentiation within the luminal epithelial cells. Intriguingly, reciprocal transplantation experiments revealed stromal, not epithelial, loss of Sfrp3 was responsible for the mammary branching/differentiation phenotypes. The authors propose the downregulation of a specific arm of Wnt signaling and suggest a Wnt2-specific defect, given the identification of an Sfrp3/Wnt2 interaction through co-immunoprecipitation experiments. The downregulation of Wnt5a and Wnt7b, in addition to Wnt2, was also noted, providing an interesting and complicated picture for

Sfrp3 regulation of both epithelial and stromal Wnt genes. The authors also present a gene expression signature derived from Sfrp3^{-/-} epithelial cells found to be differentially expressed in breast cancer vs. normal samples, yet this is largely associative in nature with respect to Sfrp3 in breast cancer.

Overall the data are sound and novel with respect to stromal Sfrp3 function on mammary branching and differentiation; however, the current manuscript would benefit from a more in depth analysis of the reported phenotypes. Additionally, the authors implicate Wnt2 as the predominant stromal Wnt ligand responsible for the Sfrp3 phenotypes, yet the mechanisms surrounding Sfrp3/Wnt2 functions are still lacking and the direct implication of Sfrp3 to breast cancer is still only associative at present. The following comments and suggestions are provided below.

- 1) The expression data in Fig. 1 are convincing; however, the current data could be strengthened by identifying the specific stromal cells responsible for Sfrp3 expression.
 - a. Can in situ hybridization or other methodologies be utilized to identify and distinguish expression in fibroblasts, adipocytes, immune cells etc.?
 - b. Are Sfrp3 levels regulated by the estrous cycle?
- 2) The increase in proliferation in Sfrp3^{-/-} mammary glands is striking. Are proliferative cells segregated from ER⁺ cells in Sfrp3^{-/-} glands?
- 3) Have the authors tried to utilize FACS analysis of subpopulations to identify subpopulation defects within the epithelial hierarchy in response to Sfrp3 depletion? This could be interesting with respect to the distribution of ML, LP, and AP populations within the luminal subset, especially since there is an accumulation of cells.
- 4) In Figure 5, the wing imaginal disc data are clear with respect to Sfrp3's ability to negatively modulate Wntless gradients and downstream targets. While very interesting, direct evidence for this in the mammary gland remains to be proven.
 - a. Can the authors demonstrate Wnt pathway activity alterations in vivo beyond Lef1 and cMYC expression changes? Although Axin2 and other downstream Wnt gene alterations are not observed, only a subset of the basal population is active, so subtle changes could be masked by evaluating the whole epithelial fraction.
 - b. The use of a transgenic Wnt reporter mouse could be useful to identify alterations in either the basal epithelial fraction or the stromal fraction. Alternatively, lentiviral Wnt reporters coupled with transplantation could be useful to show defects in vivo within the epithelium.
- 5) Is Sfrp3 interaction specific for Wnt2? Are there weaker associations with Wnt4 and Wnt7b?
 - a. Can Wnt2 gain-of-function/overexpression rescue the branching defect in Sfrp3 glands, where Wnt2 is downregulated? Preliminary evidence would suggest that ectopic Wnt2 expression results in a hypermorphic branching phenotype (Kessenbrock K, Dijkgraaf et al. Cell Stem Cell 2013). Although the current data suggest that Wnt2 functions are dysregulated in response to Sfrp3, the specific mechanism of Wnt2/Sfrp3 is still unclear from the data presented.
 - b. Does Wnt2 relay a canonical function?
- 6) Fig 4D. The white color for ZO-1 is difficult to distinguish in WT glands with eGFP overlay. Can the authors use another contrasting color?
- 7) I could not locate Table S1.

Reviewers' comments:

Reviewer #1 (Remarks to the Author):

The data demonstrating the role of SFRP3 in mammary development and the fact that it is primarily derived from the stroma is interesting, nevertheless the current manuscript leaves too many questions unanswered, and lacks solid mechanistic data.

We are glad that the reviewer finds interesting our work. We have added essential results that answer many of the questions that were not thoroughly addressed in the previous version, and provide new important mechanistic information that strengthens the message of the article.

The authors demonstrate that SFRP3 is required for proper mammary development, although there is no indication of the expression of the other SFRP genes. Is mammary development solely reliant on SFRP3? What are the relative mRNA levels of the other SFRP genes in the epithelial vs stromal cells? Furthermore what is the expression of the other SFRP members during mammary gland development at 3, 6 and 8 weeks? The observation that SFRP3^{-/-} mice results in morphological defect in the ducts, mirrors the results of a study by Gauger et al, 2012, Loss of sfrp1 promotes ductal branching in the murine mammary gland. BMC Dev Biol.

We have characterized the expression of other Sfrp genes in the mammary gland (MG) development by qPCR analysis both in the epithelial and stromal compartment (new Figure S1a). All these proteins are expressed in the MG, which complicate the analysis of the contribution of each of these proteins to the development of the MG. We obtained Sfrp1 KO and characterized its role in MG development, and we could not observe any specific phenotype. Furthermore, we have generated the double KO of Sfrp1/Sfrp3, and we observe an identical phenotype to the Sfrp3 KO in MG (not shown). We have not included this information in the article, but we would gladly share this information with the reviewer. At this point, we decided to concentrate our work in Sfrp3.

The authors use an in vivo Drosophila model to demonstrate that SFRP3 acts as a Wnt antagonist in vivo, while the data presented is quite convincing, the question remains whether in their mouse model whether SFRP3 acts as a canonical or non-canonical antagonist? The authors saw no significant changes in most classical in canonical signaling target genes, and state that it may work in a nonconventional manner. The authors saw changes in Wnt5a expression in SFRP3^{-/-} mice. Wnt5a is a known modulator of morphogenesis through ROR1, did the authors explore whether the loss of SFRP3 could lead to any changes in Wnt5a binding to ROR1 independent of an SFRP3 interaction? Or even if the loss of SFRP3 could cause an increase in the phosphorylation of Dvl? Does Wnt2 bind to a Frizzled receptor on the epithelial cells to activate Wnt signaling?

In light with the new evidence that we present in the article, we have modified the conclusions of the article substantially. Sfrp3 interact with different Wnts (2,5a, 7b) by coIP, and is very difficult at this stage to better characterize the specificity of these interactions. However, we have identified a specific activation of the canonical Wnt signaling (nuclear beta-catenin) that seems to be responsible for the over-proliferation and multilayering of the luminal compartment (new Figure 6a, b). We believe the difference of this result with our previous analysis (RNAseq) is based on the timing of the analysis. RNAseq was performed in MGs of 10-week animals when we could obtain a more significant amount of epithelial cells for the analysis. However, our analysis of Sfrp3 expression suggested a peak of expression for this protein at earlier times (3 weeks),

compared to the time in which we observe the phenotype in the epithelial compartment (6-8 week). Therefore, we decided to evaluate the effect in Wnt signaling at relatively early time points (5 weeks), and we observed a clear enrichment of nuclear beta-catenin in Sfrp3^{-/-} MGs compare to controls (figure 6a, b). We have also tested the non-canonical pathway by analyzing p-jun and p-Dvl in sorted epithelial cells in the mammary gland, and could not identify any significant difference in the Sfrp3^{-/-} MGs (not shown).

In figure 5 the authors show that differential expression of Wnt2, Wnt5a and Wnt7b in the stromal sorted cells vs the epithelial cells in SFRP3^{+/+} mice. Does the expression of these change in SFRP3^{-/-} mice?

We have analyzed the expression of Wnt2, Wnt5a and Wnt7b in the stromal vs. the epithelial sorted cells in SFRP3^{+/+} and SFRP3^{-/-} mice. We have not identified significant differences in the expression of these Wnts between control and Sfrp3 KO animals in epithelial and stromal sorted cells. In the revised version of the manuscript we have eliminated the results showing differences between the different Wnts in control and Sfrp^{-/-}

The immunoprecipitation data is interesting, demonstrating that the NTR is required for secretion. The authors state SFRP3 "very likely binds through its CRD". This should be confirmed experimentally through co-immunoprecipitation.

We are glad the reviewer suggested this experiment. We have performed this experiment (new figure 6h). Contrary to what we initially hypothesized, the interaction analysis experiment of the different sfrp3 domains with the Wnts tested in this study shows that the interaction is explicitly produced by the NTR domain (Figure 6h). Although the NRT domain of sfrp3 binds to all the Wnts analyzed, the interaction is particularly strong with Wnt2, similar to the observation with the complete protein by CoIP (Figure 6g). This result was unexpected for us, but the specific interaction with Wnts through the NRT domain of a Sfrp protein was previously described for Sfrp1 (Uren, 2000; Lopez-Rios J, 2008).

The data demonstrate that loss of SFRP3 results in a transcriptomic profile which exhibits a more cancer-associated profile, although they were not able to generate any mammospheres in SFRP null mice and did not report any tumour development in the mouse model. Do tumours develop in these mice in the longer term?

We have characterized the effect of sfrp3 in tumor development in the long term, and we found no significant difference compared to controls (not shown). However, it has been previously described alterations in genes that predispose the appearance of cancer, are not sufficient to induce that transformation, but require additional mutations that generate a sensitized background. In order to assess the tumorigenicity capacity of sfrp3, we crossed sfrp3^{-/-} mice with the MMTV-Neu^{+/-} (MMTV-ErbB2), that generate tumors which are classified as human luminal type carcinomas (figure 7 e-g). Interestingly, we observed the formation of preneoplastic lesions in the sfrp3^{-/-} MMTV-Neu mammary gland samples which are phenotypically similar to the MMTV-Neu premalignant lesions described before (figure 7e) (van de Vijver MJ, 2002). Furthermore, we identified that lack of sfrp3 in MMTV-Neu animals induces the precocious appearance of premalignant lesions (figure 7f), suggesting a role for Sfrp3 as a new marker for breast cancer susceptibility. The analysis of the Sfrp3^{-/-}; MMTV-Neu^{+/-} mammary gland tumors by confocal microscopy showed that the tissues present the characteristic features of the Sfrp3^{-/-} mammary ducts with beta casein accumulated in the lumen of ducts and hyperproliferative luminal cells (figure 7g).

Specific Comments:

1. The title doesn't reveal the subject content. It gives an impression that the study is going to be on normal mammary gland development. Abstract describes both the cancer and mammary gland development.

We agreed with the reviewer, and we have modified the title of the article to include the role of Sfrp3 in cancer

2. Excellent animal studies. However, clear hypothesis, aims and the connectivity with relevant experimental model is missing.

We are glad that the reviewer finds our work remarkable. We have added essential results that addressed many of the questions that remained open in the previous version and provided new important mechanistic information that strengthens the message of the article.

3. It is known sFRPs are downregulated/dysregulated in cancers. The experimental approaches should have been focused towards mammary gland development/milk secretion/ production rather than reporting for tumour susceptibility.

As mention before, now we provide additional information regarding the connection with cancer susceptibility

4. Sfrp3^{+/+} status will be interesting in section "loss of Sfrp3 leads to structural defects of epithelial ducts and precocious differentiation of mammary gland"

We are not sure we have understood the reviewer's question. In figure 2 that refers to the section "Loss of Sfrp3 leads to structural defects of epithelial ducts and precocious differentiation of mammary gland" the information of the control Sfrp3^{+/+} is included, to compare it with the Sfrp3^{-/-}

5. The method authors used did not completely block estrogen production. Other sources of hormone production should be kept in mind. Did not see any change in estrogen and progesterone in the serum indicating the presence of the hormones from other sources. Cannot comment on effect of hormones on sfrp3 unless it is totally abolished.

We agreed with the reviewer than the ovariectomy does not entirely block estrogen production, but we observed that in the absence of ovarian hormones Sfrp3^{-/-} ducts do not elongate, like Sfrp3^{+/+} ducts, and do not show any alteration in branching rate or structural organization. Therefore, this result demonstrates that lack of Sfrp3 is not sufficient to induce the phenotype described in the absence of the hormones produces by the ovary. We have remarked this critical point in the new version of the article. Also, we measured the serum levels of oestrogen and progesterone in Sfrp3^{+/+} and Sfrp3^{-/-}, and we do observe any significant change in their levels, precluding the possibility that the branching phenotype of Sfrp3^{-/-} mice might be due to an increase in hormone levels as a secondary effect.

6. Although drosophila is an excellent model to study morphogens, it may not be the biologically closer model to mice, better approaches using cell line or breast cancer cells would have been more appropriate in section "Sfrp3 controls mammary gland development thought the modulation of Wnt signalling"

With all due respect, we do not agree with the reviewer's comments. Our experiments in *Drosophila* nicely complement and reinforce the observations in the mice mammary gland. *Drosophila* imaginal disc development is an excellent model system to study the role of different morphogens, such as Wingless, which have been meticulously characterized throughout countless studies in the last decades. It also has the significant advantage of being significantly simpler in terms of redundancy of morphogen expression compared to more complex systems such as the mammary gland. Moreover, taking into account that the study model is based on crosstalk between different tissues (stromal and epithelial), we believe it is more informative to model this signaling in relatively simpler systems, but that keep the complexity and plasticity of real tissue. On top of that, we have also verified our hypothesis in experiments performed in the mammary gland, which reinforce and confirm the conclusions derived from the work in *Drosophila*.

7. What will be the status of WNTs in sFRP3^{+/+} or the wild type in section "Sfrp3 controls mammary gland development thought the modulation of Wnt signaling?"

We found a significant down-regulation of *Wnt2*, *Wnt5a* and *Wnt7b* mRNA levels in adult *Sfrp3*^{-/-} mammary glands compared to controls *Sfrp3*^{+/+}, which is consistent with the decrease we observed in pregnant mice. However, since we have not identified significant differences in the expression of *Wnt2*, *Wnt5a* and *Wnt7b* mRNA between control and *Sfrp3* KO animals in epithelial and stromal sorted cells, we decided to eliminate these results from the revised version of the manuscript.

8. The authors showed the interaction in hek293 cells and relating it to mammary stroma. The best approach would be to do IPs-assay on the tissues of mammary gland. They saw a decrease in WNT2 mRNA in sFRP3^{-/-} knock out mice in section "Sfrp3 controls mammary gland development thought the modulation of Wnt signalling"

We agree with the reviewer that this would be the best option. However, despite our multiple attempts to obtain an antibody against *Sfrp3* that detects the endogenous protein by IP, we do not have appropriate tools to perform this experiment. Furthermore, the antibodies we have tested are not capable of recognizing *Wnt2* for IP in the mammary gland. Therefore our best approach is for the moment to use systems with exogenously expressed tagged forms of *Sfrp3* and *Wnts*.

9. Though the authors did not observe tumorigenic properties they cannot conclude sfrp3 does not act as a tumor suppressor gene. This should be studied in a mice tumor model targeting cells expressing sfrp^{+/+} in section "Loss of Sfrp3 leads to alteration in the expression level of genes associated with cancer"

As mention before, we have characterized the effect of *Sfrp3* in tumor development in the long term, and we found no any significant difference compared to controls (not shown) However, it has been previously described alterations in genes that predispose the appearance of cancer, are not sufficient to induce that transformation, but require additional mutations. In order to assess the tumorigenicity capacity of *sfrp3*, we crossed *sfrp3*^{-/-} mice with the MMTV-*Neu*^{+/-} (MMTV-*ErbB2*), which generate tumors which are classified as human luminal type carcinomas (van de Vijver MJ) (new figure 7e-g). Interestingly, we observed the formation of preneoplastic lesions in the *sfrp3*^{-/-}; MMTV-*Neu* mammary gland samples which are phenotypically similar to the MMTV-*Neu* premalignant lesions described before (van de Vijver MJ). Interestingly, we also identified that lack of *sfrp3* in MMTV-*Neu* animals induces the appearance of tumorigenic lesions in the mammary gland

at earlier times than in control MMTV-Neu samples (Figure 7f). These lesions also present the phenotypic characteristics described before for *Sfrp3*^{-/-} mammary ducts including hyperproliferation and multilayering of luminal cells (Figure 7g).

10. The authors jumped into human breast cancer gene data set to correlate the mice sfrp3^{-/-} NGS status. Non-tumorigenic expression cannot be compared with tumorigenic expression as the receptors and ligands interaction may vary in section "Loss of Sfrp3 leads to alteration in the expression level of genes associated with cancer"

I am afraid we did not explain with sufficient clarity the experiment in the previous version of the article. We performed an NGS analysis of sorted mammary gland epithelial cells (CD49^{low}EpCAM^{high}) of *Sfrp3*^{-/-} and *Sfrp3*^{+/+} to identify transcriptomic changes in the epithelial cell populations of mutant mammary glands pointing to tumorigenic susceptibility. In the new version, this is more clearly explained

11. In section "Loss of Sfrp3 leads to alteration in the expression level of genes associated with cancer" above should be studied in a tumor model concluding for anticancer properties.

We thank the reviewer for the suggestion, we have included in the current version the breast tumor model MMTV-Neu (MMTV-ErbB2), tumors, which are classified to produce human luminal type carcinomas (van de Vijver MJ) (new figure 7e-f). Interestingly, we observed the formation of preneoplastic lesions in the *sfrp3*^{-/-}; MMTV-Neu mammary gland samples which are phenotypically similar to the MMTV-Neu premalignant lesions described before (van de Vijver MJ, 2002). Interestingly, we also identified that lack of *sfrp3* in MMTV-Neu animals induces the appearance of tumorigenic lesions in the mammary gland at earlier times than in control MMTV-Neu breast samples (Figure 7f). These lesions also present the phenotypic characteristics described before for *Sfrp3*^{-/-} mammary ducts including hyperproliferation and multilayering of luminal cells (Figure 7g)

12. In discussion, the section suggest that stromal SFRP3 binds to stromal Wnt2 to maintain Wnt2 normal expression level needs to be proven.

Given the recently observed results, our conclusion of the results varies slightly with the previous hypothesis. Our latest results show that *Sfrp3* would bind *Wnt2*, and perhaps other Wnts, functioning as a transporter and diffuser of these morphogens, and not as an inhibitor. The expression of *Sfrp3* would thus reduce the levels of the Wnt proteins where they are more concentrated (for example *Wnt2* in the stromal-epithelial boundary), and thus reducing their signaling. The deletion of *Sfrp3* would cause *Wnt2* (and other Wnts) to accumulate in the stromal-epithelial boundary (Figure 6d), thereby inducing the anomalous signaling that produces precocious branching and multilayering in the luminal compartment.

General Comment: Experimental models should be specific to the questions. The aim should be to address the mammary gland development. Instead, authors are trying to relate mammary gland development during pregnancy to cancer, addressed sfrp 3 interaction with wnts in cell lines but not clearly addressed sFRP3/- knock out mice scenario in puberty or pregnancy or in absence of hormones. Showing evidence for targeting sFRP3 stroma cells would have been a better approach

This work aimed to focus on addressing the role of Sfrp3 in mammary gland development, and its connection with breast cancer susceptibility. For this, we used a knockout mouse for the sfrp3 gene as the base model. However, we have also used other model systems that, given their relative simplicity, have given us essential information that is very relevant in the function of this gene at the cellular and molecular level. For example, the Drosophila model has the significant advantage of expressing relatively few Wnt genes, which is very beneficial when it comes to identifying mechanisms of action in this signaling pathway. With the new information we provide in this new version of the manuscript, we firmly believe that we present enough evidence of the role of SFRP3 in the regulation of the mammary gland development and cancer as to be emerging interesting for the community of readers of Nature Communications.

Reviewer #2 (Remarks to the Author):

Bernascone and colleagues report the study of Sfrp3, a member of a secreted glycoprotein family, as a mediator of cross talk between stroma and epithelium during mammary gland development. Per request from the editors, my review will focus primarily on the part of the study conducted in Drosophila but will also touch upon what I perceive as major weaknesses in the whole report. In Drosophila, the authors overexpressed myc-tagged Sfrp3 in half of the larval wing discs. The GAL4 driver used would result in expression in the columnar epithelial cells. This 'interferes with the symmetrical gradient of Wingless' (Drosophila Wnt1) and reduced the expression of Wg target Sen. The authors show that Notch signaling is not affected in the same system, indicating some specificity. While the effect on Sen supports the conclusion that Sfrp3 overexpression inhibits Wg, I cannot see the effect on Wg gradient that the authors refer to. The staining still looks quite symmetrical across the D/V boundary to me, and if anything, I see reduced Wg in the P compartment where Sfrp3 is expressed. To be more convincing, the authors need to quantify Wg levels by quantitative image analysis such as a line scan across the D/V boundary. The reduction of Wg by Sfrp3, if real, is puzzling because the literature cited by the authors (ref 11-12) suggest that Sfrp3 homologs inhibit Wnt signaling (by direct binding to Wnt proteins), not Wnt production. So, it is odd that Sfrp3 in wing discs would reduce the level of Wg. The effect could be indirect, as in blocking the proliferation and reducing the number of Wg-producing cells, for example, and may or may not involve inhibition of Wg signaling.

We agree with the reviewer that Wg staining seems to be reduced in the P compartment, but symmetrical in the D/V boundary. Following the reviewer's suggestions, we have analyzed Wg levels by quantitative image analysis with a line scan across the D/V boundary. We found that Wg is not reduced in total levels, but the peak at the DV is reduced (New Figure 5a, b), which indeed correlated very well with the current hypothesis we have of Sfrp3 function.

In the last months, we performed different experiments signaling, and we obtained the following conclusions:

1- In addition to Sens, we have also characterized Distalless (dll), a Wg target that response to low Wg concentrations. We observed that Sfrp3 expression does not inhibit dll, but on the contrary, it broadens the dll territory in the P compartment (New Figure 5f). Thus, while the effect on Sen seems to support the conclusion that Sfrp3 overexpression inhibits Wg, the effect on dll suggests that Sfrp3 expression might induce the transport and diffusion of Wg, expanding Wg territory but reducing its concentration at the D/V boundary. This effect of Sfrp3 would inhibit Sens (which is a high-concentration responsive Wg-effector) but increase dll (which is a low-concentration responsive effector).

2- In order to prove this role of Sfrp3 in the transport and diffusion of Wg, we used a membrane-bound form of Wg (NRT-wg), described before to allow persistent expression of

relevant target genes, suggesting therefore that the spread of Wingless is dispensable for patterning and growth (Alexandre et al., Nature 2014). Importantly, the expression of Sfrp3 in the posterior compartment did not affect Sens (Figure 5g, h) or dll (not shown) in NRT-Wg wing imaginal discs, demonstrating that Sfrp3 is not a repressor or inhibitor of Wg, but more likely a transporter of Wg that allow the spreading of this morphogen.

What is even more puzzling is the authors' finding that Sfrp3^{-/-} tissues show decreased expression of some Wnt proteins (Fig.5), opposite of what you expect from the Drosophila data where overexpression of Sfrp3 decreased Wg. Again, the authors' finding is contrary to the literature that Sfrp proteins, as secreted extracellular proteins, would affect Wnt function outside of the cell but not necessarily Wnt expression.

We agree with the reviewer that Sfrp proteins, as secreted extracellular proteins, would affect Wnt function outside of the cell but not necessarily Wnt expression. Indeed, we do not believe at this point that Sfrp3 function as an inhibitor but rather as extracellular transporter. As mention before, we found no difference in Wg expression with Sfrp3 expression, but a decrease of the concentration at the D/V boundary causing alterations in the Wg target genes Sens and dll (figure 5).

Related, the authors conclude that secreted Sfrp3 binds to Wnt2 (outside the cell) to maintain Wnt2 expression in the same cells. Is there precedent for Wnt signaling acting to maintain Wnt expression? If Wnt2 signaling is really affected, why are targets of Wnt signaling not altered and there was no change in beta-catenin nuclear localization in Sfrp3^{-/-} cells (pg. 13)? The authors discuss Sfrp proteins as inhibitors of Wnt signaling. But if Sfrp3^{-/-} mutation reduces Wnt2 levels, wouldn't the wild type then be an activator of Wnt2?

We agree with the reviewer that if Wnt2 signaling is affected, targets of Wnt signaling should also be altered in Sfrp3^{-/-} cells, such as beta-catenin nuclear localization. Indeed, we have analyzed the beta-catenin nuclear localization at earlier times (5weeks) when Sfrp3 levels are higher in the MG of control animals (Sfrp3^{+/+}), and we found a significant increase in nuclear beta-catenin localization in Sfrp3^{-/-} MG (Figure 6a, b) that we believe reflects the increased accumulation of Wnt2 (and perhaps other Wnts) at the Stromal-epithelial boundary in Sfrp3^{-/-} MG (Figure 6d). At this point, we can not determine how the expression levels of Wnt2, 5a, and 7b are regulated, and why they are modified in the KO of Sfrp3. It would be very speculative for us to indicate that it is by self-regulation. In addition, we could not observe significant differences when we analyzed sorted epithelial and stromal populations, only in the de-epithelialised fat pads (Figure 6e). We have decided to eliminate these data (Wnt2, 5a and 7b in whole MG) in the revised version of the manuscript until we fully understand this regulation.

The major weakness in this report is the use of 'stromal' when it is really 'systemic'. For example, in transplantation experiments in Fig. 4, mutant epithelium developed normally when placed into wild type stroma but wild type epithelium developed abnormally when placed into mutant stroma. The authors conclude that Sfrp3 in the stroma directs normal development, but it is not just the stroma but the whole rest of the mouse that bears the genotype of the stroma in each case. So it could be contribution from another tissue and not the stroma that is important. In another example, Sfrp3^{-/-} stroma is shown to express reduced Wnt2 compared to wild type stroma (Fig. 5F). The authors conclude that Sfrp3 function is 'cell compartment

mautonomous' in directing Wnt 2 expression. Again, it is not just the stroma that is mutant but the whole mouse is. It could be Sfrp3 function in another tissue that is responsible for Wnt expression in the stroma. To conclude that stromal Sfrp3 is important for epithelial development, tissue-specific knock down/out of Sfrp3 in the stroma is needed.

Following the reviewer's suggestions, we have tested the “stromal vs. systemic” contribution of Sfrp3 in the phenotype of Sfrp3^{-/-} in MG. Instead of a tissue-specific knockdown/out of Sfrp3 in the stroma that we did not have, or did not work (knock-down), we designed a different strategy. We used a transplantation experiment using control Sfrp3^{+/+} as a donor epithelial transplanted into an Sfrp3^{-/-} recipient fat pad and altogether transplanted in a control nude animal (and thus Sfrp3^{+/+}). If the role of Sfrp3 is systemic, we would expect to have no multilayering and hyperbranching phenotype in these MGs. However, we obtain a phenotype that is very similar to the phenotype observed in the transplantation of a wild type epithelium placed into mutant stroma (Figure S3f).

In sum, the developmental phenotype of Sfrp3^{-/-} mice are extremely interesting. But the data from follow up analyses to get at the mechanism fall short of supporting the conclusions.

We are glad that the reviewer finds our work "extremely interesting. As mention before, we have added essential results that addressed many of the questions that remained open in the previous version and provided new important mechanistic information that strengthens the message of the article.

Minor point on data presentation; it is potentially confusing to show y-axis values that are so different in Fig. 5 D, E and F, for the same gene. I understand that large values result from very small levels in one sample used to normalize (like dividing by zero), but it would be more informative to normalize relative to a single sample for each gene (e.g. use whole gland Wnt 2 in wild type to normalize all other Wnt2 signals). That would allow comparison across time points and also across tissues.

We agreed, but we used different normalizers in the different experiments (ppia, b-actin, GAPDH, etc) and they are difficult to compare. In addition, we have eliminated some of the data in previous figure 5 (new figure 6), as discussed before.

Reviewer #3 (Remarks to the Author):

This is a very interesting study by Bernascone I et al. that describes a novel role for the secreted Wnt inhibitor, Sfrp3, in regulating branching morphogenesis and differentiation of the mammary epithelium through the stromal-epithelial crosstalk. The authors provide convincing expression data that Sfrp3 predominantly resides in the stroma of the mammary gland and is downregulated upon pregnancy. Genetic loss-of-function of Sfrp3 prompted accelerated ductal extension and branching during puberty and precocious alveolar development in the adult. These phenotypes were accompanied by alterations in luminal polarity, multicellular accumulation, and defects in differentiation within the luminal epithelial cells. Intriguingly, reciprocal transplantation experiments revealed stromal, not epithelial, loss of Sfrp3 was responsible for the mammary branching/differentiation phenotypes. The authors propose the downregulation of a

specific arm of Wnt signaling and suggest a Wnt2-specific defect, given the identification of an Sfrp3/Wnt2 interaction through co-immunoprecipitation experiments. The downregulation of Wnt5a and Wnt7b, in addition to Wnt2, was also noted, providing an interesting and complicated picture for Sfrp3 regulation of both epithelial and stromal Wnt genes. The authors also present a gene expression signature derived from Sfrp3^{-/-} epithelial cells found to be differentially expressed in breast cancer vs. normal samples, yet this is largely associative in nature with respect to Sfrp3 in breast cancer.

Overall the data are sound and novel with respect to stromal Sfrp3 function on mammary branching and differentiation; however, the current manuscript would benefit from a more in depth analysis of the reported phenotypes. Additionally, the authors implicate Wnt2 as the predominant stromal Wnt ligand responsible for the Sfrp3 phenotypes, yet the mechanisms surrounding Sfrp3/Wnt2 functions are still lacking and the direct implication of Sfrp3 to breast cancer is still only associative at present.

We are glad that the reviewer finds our work sound and novel. We have added essential results that answer many of the questions that were not thoroughly addressed in the previous version, and provide new important mechanistic information that strengthens the message of the article.

The following comments and suggestions are provided below.

- 1) The expression data in Fig. 1 are convincing; however, the current data could be strengthened by identifying the specific stromal cells responsible for Sfrp3 expression.
 - a. Can in situ hybridization or other methodologies be utilized to identify and distinguish expression in fibroblasts, adipocytes, immune cells etc.?*
 - b. Are Sfrp3 levels regulated by the estrous cycle?**

We agreed, and following reviewer's suggestions we have characterized the source of Sfrp3 in the stromal cells. Sfrp3 is expressed by fibroblast almost exclusively as evidenced by the qPCR analysis of isolated cell populations (fibroblast, Adipocytes and Cd45⁺ immune cells) (New Figure S1c). Despite our multiple attempts to obtain an antibody against Sfrp3 that detects the endogenous protein by IP, we could not generate the appropriate tools to perform this experiment. Similarly, ISH in fat pads are extremely difficult to perform, and we could not get a reliable result.

We also tested the levels of Sfrp3 along the oestrous cycle. As shown in figure S5c, we could not observe significant differences of Sfrp3 along the cycle.

- 2) The increase in proliferation in Sfrp3^{-/-} mammary glands is striking. Are proliferative cells segregated from ER⁺ cells in Sfrp3^{-/-} glands?*

Indeed, we agreed that the increase in proliferation in Sfrp3^{-/-} mammary glands is striking. We have evaluated the ER⁺ vs. ER⁻ cell populations and found that Sfrp3^{-/-} MG ducts present a dramatic decrease in the ER⁺ cell population that suggest that proliferative cells are segregated from the ER⁺ population (New figure 3d, e).

- 3) Have the authors tried to utilize FACS analysis of subpopulations to identify subpopulation defects within the epithelial hierarchy in response to Sfrp3 depletion? This could be interesting with respect to the distribution of ML, LP, and AP populations within the luminal subset, especially since there is an accumulation of cells.*

We have indeed characterized the Sfrp3^{-/-} MGs to identify alteration in the cellular subpopulations. Interestingly, we observed an increase in the NaPi⁺ alveolar subpopulation

(Figure 3h) which correlates with the increase of beta-casein production (and other milk-associated proteins) in the *Sfrp3*^{-/-} MGs. Also, we observed an increase in the mature luminal subpopulation as evidenced by the expression of Troma/K8 staining (Figure 3c).

4) In Figure 5, the wing imaginal disc data are clear with respect to Sfrp3's ability to negatively modulate Wingless gradients and downstream targets. While very interesting, direct evidence for this in the mammary gland remains to be proven.

a. Can the authors demonstrate Wnt pathway activity alterations in vivo beyond Lef1 and cMYC expression changes? Although Axin2 and other downstream Wnt gene alterations are not observed, only a subset of the basal population is active, so subtle changes could be masked by evaluating the whole epithelial fraction.

b. The use of a transgenic Wnt reporter mouse could be useful to identify alterations in either the basal epithelial fraction or the stromal fraction. Alternatively, lentiviral Wnt reporters coupled with transplantation could be useful to show defects in vivo within the epithelium.

As mention before (see answers to reviewer 2), as a result of the new evidence we have modified our hypothesis of *Sfrp3* function. Indeed, we do not believe at this point that *Sfrp3* function as an inhibitor but rather as extracellular transporter that mediates the diffusion of Wnts in specific locations. *Sfrp* proteins, as secreted extracellular proteins, would affect Wnt function outside of the cell but not necessarily Wnt expression. We found no difference in Wg distribution upon *Sfrp3* expression, but a decrease of the concentration at the D/V boundary causing alterations in the Wg target genes *Sens* and *dll* (New Figure 5). We have also tested this hypothesis in the MG. Indeed, we have analyzed the beta-catenin nuclear localization at earlier times (5weeks) when *Sfrp3* levels are higher in the MG of control animals (*Sfrp3*^{+/+}). New evidence shows that nuclear beta-catenin is increased in the *Sfrp3*^{-/-} MG (Figure 6a, b), that we believe it reflects the increased accumulation of Wnt2 (and perhaps other Wnts) at the Stromal-epithelial boundary in *Sfrp3*^{-/-} MG (Figure 6e).

5) Is Sfrp3 interaction specific for Wnt2? Are there weaker associations with Wnt4 and Wnt7b?

a. Can Wnt2 gain-of-function/overexpression rescue the branching defect in Sfrp3 glands, where Wnt2 is downregulated? Preliminary evidence would suggest that ectopic Wnt2 expression results in a hypermorphic branching phenotype (Kessenbrock K, Dijkgraaf et al. Cell Stem Cell 2013). Although the current data suggest that Wnt2 functions are dysregulated in response to Sfrp3, the specific mechanism of Wnt2/Sfrp3 is still unclear from the data presented.

We cannot conclusively demonstrate that the effect we see in *Sfrp3*^{-/-} animals in MG is due exclusively to the interaction of *Sfrp3* with Wnt2. However, previous evidence would suggest that ectopic Wnt2 expression results in a hypermorphic branching phenotype (Kessenbrock K, Dijkgraaf et al. Cell Stem Cell 2013), as is indicated by the reviewer. We observe an accumulation of Wnt2 in the stromal/epithelial boundary in *Sfrp3*^{-/-} MGs (figure 6e) that might induce the overproliferation and hypermorphic branching phenotype that we identified in our mutants.

b. Does Wnt2 relay a canonical function?

As mention above, new evidence shows that nuclear beta-catenin is increased in the *Sfrp3*^{-/-} MG (Figure 6a, b), that we believe it reflects the increased accumulation of Wnt2 (and perhaps other Wnts) at the Stromal-epithelial boundary in *Sfrp3*^{-/-} MG (Figure 6d).

6) Fig 4D. The white color for ZO-1 is difficult to distinguish in WT glands with eGFP

overlay. Can the authors use another contrasting color?

We agree, we have modified this pictures (new figure 4d), and now ZO1 is showed in red.

7) I could not locate Table S1.

We have uploaded the Table S1 again

Reviewers' Comments:

Reviewer #1:

None

Reviewer #2:

Remarks to the Author:

The revision by Martin-Belmonte and colleagues addresses all my concerns from the previous round of review. I recommend its publication in Nature Communications.

Reviewer #3:

Remarks to the Author:

The authors have responded to most of the concerns of the original reviews and added a significant amount of new data. I think this manuscript is now suitable for publication.

Reviewers' comments:

Referee #1 provided confidential comments to the Editor but here, they stated: "I am happy with the revision. The manuscript is suitable for publication in the present form."

Reviewer #2 (Remarks to the Author):

The revision by Martin-Belmonte and colleagues addresses all my concerns from the previous round of review. I recommend its publication in Nature Communications.

Reviewer #3 (Remarks to the Author):

The authors have responded to most of the concerns of the original reviews and added a significant amount of new data. I think this manuscript is now suitable for publication.

We are glad that the reviewers find interesting our work and recommend its publication in Nature Communications.